# LitLLMs, LLMs for Literature Review: Are we there yet?

**Shubham Agarwal \***
*ServiceNow Research, Mila - Quebec AI Institute, HEC Montreal*

**Gaurav Sahu \***
*ServiceNow Research, University of Waterloo*

**Abhay Puri \***
*ServiceNow Research*

**Issam H. Laradji**
*ServiceNow Research, University of British Columbia*

**Krishnamurthy DJ Dvijotham**
*ServiceNow Research*

**Jason Stanley**
*ServiceNow Research*

**Laurent Charlin**
*Mila - Quebec AI Institute, HEC Montreal, Canada CIFAR AI Chair*

**Christopher Pal**
*ServiceNow Research, Polytechnique Montreal, Mila - Quebec AI Institute, Canada CIFAR AI Chair*

*\* Equal contribution*

**Reviewed on OpenReview:** *https://openreview.net/forum?id=heeJqQXKg7*

## Abstract

Literature reviews are an essential component of scientific research, but they remain time-intensive and challenging to write, especially due to the recent influx of research papers. This paper explores the zero-shot abilities of recent Large Language Models (LLMs) in assisting with the writing of literature reviews based on an abstract. We decompose the task into two components: (1) Retrieving related works given a query abstract and (2) Writing a literature review based on the retrieved results. We analyze how effective LLMs are for both components. For retrieval, we introduce a novel two-step search strategy that first uses an LLM to extract meaningful keywords from the abstract of a paper and then retrieves potentially relevant papers by querying an external knowledge base. Additionally, we study a prompting-based re-ranking mechanism with attribution and show that re-ranking doubles the normalized recall compared to naive search methods while providing insights into the LLM's decision-making process. In the generation phase, we propose a two-step approach that first outlines a plan for the review and then executes steps in the plan to generate the actual review. To evaluate different LLM-based literature review methods, we create test sets from arXiv papers using a protocol designed for rolling use with newly released LLMs to avoid test set contamination in zero-shot evaluations. We release this evaluation protocol to promote additional research and development in this regard. Our empirical results suggest that LLMs show promising potential for writing literature reviews when the task is decomposed into smaller components of retrieval and planning. Particularly, we find that combining keyword-based and document-embedding-based search improves precision and recall during retrieval by 10% and 30%, respectively, compared to using either of the methods in isolation.

Further, we demonstrate that our planning-based approach achieves higher-quality reviews by minimizing hallucinated references in the generated review by 18-26% compared to existing simpler LLM-based generation methods. Our project page including a demonstration system and toolkit can be accessed here: https://litllm.github.io.

# 1 Introduction

Writing a literature review—finding, citing, and contextualizing relevant prior work—is a fundamental scientific writing requirement. When writing manuscripts, scientists must situate their proposed ideas within the existing literature. Writing a good literature review is a complex task which can be broken down into two broad sub-tasks: **1)** Finding relevant papers and **2)** Generating a related work section to discuss the proposed research given prior works. This challenge is further amplified in fields such as machine learning, where thousands of relevant papers appear every month on arXiv alone.[1] We explore the utility and potential of large language models (LLMs), in combination with retrieval mechanisms, to assist in generating comprehensive literature reviews for scientific papers.

Specifically, we investigate using LLMs to generate a paper's related work section based on its abstract. We use the term abstract loosely, not necessarily to refer to the actual abstract of the paper but rather to a textual passage that captures a concise summary of the paper's key contributions and scope. Using the abstract as input allows our system to target the central ideas of the paper without requiring the complete manuscript, which is often continuously evolving in the early stages of writing. While our experiments focus on using the abstract, our framework is designed to be flexible. It can use the entire manuscript as it evolves, albeit at a higher computational cost and the need to use models that support longer context windows. This approach provides valuable early-stage insights for authors seeking preliminary references to shape their work, with the capacity to seamlessly incorporate additional information as the manuscript develops.

The architecture of our framework is illustrated in Figure 1, where we further decompose each of the two above tasks into two subtasks. In this work: **1)** We introduce an LLM-based approach to retrieve relevant papers, where we first extract the keywords from an abstract or research idea paragraph using an LLM and then feed these keywords to a keyword-based search tool — we experiment with Google search and Semantic Scholar. We optionally also transform the abstract or idea into an embedding and use an embedding-based search procedure. **2)** We then employ a prompting-based approach to rank the set of retrieved candidate papers based on their relevance to the query abstract, also requiring the LLM to attribute the relevance to specific excerpts in the candidate papers. We explore multiple re-ranking and aggregation strategies. **3)** To generate the literature review, we select top-$k$ papers from the ranked list and prompt the LLM to generate the related work section based on the query abstract and the abstracts of the selected papers. **4)** Additionally, we examine the effectiveness of providing a writing plan to the LLM that specifies which papers to cite at various points in the literature review. These plans can be generated entirely by the LLM, by the user, or a combination of the two. These plans serve as an intermediate representation giving the user more control over the organizational structure of the literature review.

The complete framework involves multiple innovations, where we use LLMs in multiple ways, namely for generating search queries, re-ranking search results, and attribution. We summarize the main contributions of our work as follows:

- To answer the key question that our paper poses, we present a data collection protocol and multiple instances of using it to collect arXiv papers. Critically, our protocol is based on using the most recent month of arXiv papers in a rolling manner with the goal of avoiding test-set contamination when evaluating the most recent LLMs for literature review-related tasks. We then use this protocol to perform extensive retrieval and literature review generation experiments. We release both our datasets and our code to the community.

- We propose a novel LLM-based pipeline for the task of interactive literature review writing, which we decompose into two distinct components: retrieval and generation. This also facilitates more controlled

---

[1]E.g. over 4,000 ML papers were submitted to arXiv in October 2024: https://arxiv.org/list/cs.LG/2024-10

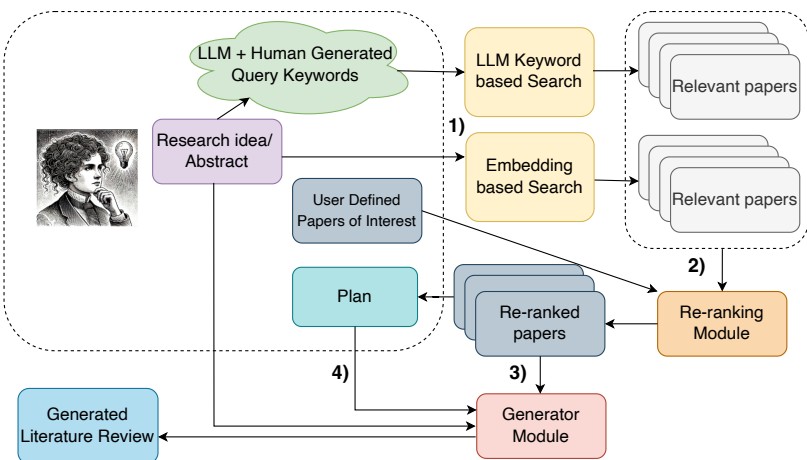

Figure 1: A schematic diagram of our framework, where: 1) Relevant prior work is retrieved using keyword and embedding-based search. 2) LLMs re-rank results to find the most relevant prior work. 3) Based on these papers and the user abstract or idea summary, an LLM generates a literature review, 4) optionally controlled by a sentence plan.

studies investigating alternative LLM-based approaches for these sub-tasks. Our experiments focus on evaluating fully automated variants of these sub-tasks, but our framing of the problem and our proposed solutions are easily integrated into scenarios where human users interact with systems that assist them. To the best of our knowledge, our decomposition of the problem into sub-tasks, along with our proposed solutions to them, and our framing of this assistive scenario is novel.

- We make multiple contributions that improve the retrieval phase of the two-step process. First, we propose and evaluate a novel strategy that first uses an LLM to extract meaningful keywords from the abstract of a paper and then queries different types of external sources to retrieve potentially relevant papers. Second, we compare and combine the aforementioned LLM-generated keyword search techniques with document embedding-based retrieval methods. Third, we propose and examine a wide variety of search re-ranking techniques. Among these, we propose a novel prompting for attribution approach, which we find to empirically improve the relevance of retrieved literature while also improving reliability, providing insights and improving transparency in the decision-making process of LLMs when used to rerank results. Going even further, we examine debate-prone LLMs for aggregating and re-ranking keyword search and embedding search results. Our experiments show that combining these ideas improves precision and normalized recall by 10% and 30%, respectively, compared to standard retrieval methods.

- For text generation, we propose and examine a plan-based retrieval augmented approach to writing literature reviews. By using a plan and conditioning on retrieved context, we provide a user greater control over generated content, and our experiments show that this approach can improve the quality of the generated literature reviews substantially. We evaluate the approach using automated metrics and human assessments and show that our method generates higher-quality reviews as measured by ROUGE scores and as assessed by human evaluation. Our approach also reduces hallucinations by 18–26%.

## 2 Related Work

We decompose the literature review task into two key sub-tasks: identifying relevant papers and generating the final related work section. This decomposition enhances the likelihood that the LLMs can effectively accomplish the task. We now discuss the relevant literature pertinent to both aspects of the process.

### 2.1 Ranking and Retrieval

Traditional methods for information retrieval rely on techniques like TF-IDF and BM25 to identify documents that are semantically similar to a given query. More recently, dense vector representations obtained through models like Sentence-BERT (Reimers & Gurevych, 2019) have been shown to improve retrieval accuracy by encoding both query and documents into an embedding space where semantic similarity can be readily computed. The initial retrieval stage often results in a large set of candidate documents, which then need to be re-ranked to obtain an ordered list based on relevance.

Recent efforts have explored the application of proprietary and open-source LLMs for ranking (Sun et al., 2023; Ma et al., 2023; Pradeep et al., 2023a;b), where the LLM is passed a combined list of passages directly as input and prompted to rank them based on a criteria. Notably, only top-k candidates are passed as input to the LLM for re-ranking (Zhang et al., 2023). In our work, we instruct an LLM to output an ordering of the different candidate papers (e.g. [3] > [8] > [6]) in descending order based on the relevance to the user-provided abstract. Although existing re-ranking methods improve the ordering of candidate papers, they do not provide explicit justification for the relative rankings assigned. To improve the reliability of the system and offer a clear explanation for the model's choices, our methodology also incorporates attribution capabilities, allowing us to identify specific textual elements contributing to relevance scores for different candidates. Among works exploring the attribution capabilities in LLMs, Yue et al. (2023) focuses on automatically evaluating whether generated statements are fully supported by cited references. Cohen-Wang et al. (2024) present ContextCite, a method for attributing model generation to context, which can be applied on top of any existing language model to help verify generated statements, improve response quality, and detect poisoning attacks. Gradient-based techniques, such as Integrated Gradients (Sundararajan et al., 2017) and Saliency Maps (Simonyan et al., 2014), measure the contribution of each input token by computing gradients of the output with respect to input features; advances like SmoothGrad (Smilkov et al., 2017) and DeepLIFT (Shrikumar et al., 2017) improve these methods by reducing noise and enhancing accuracy. Perturbation techniques involve modifying or occluding parts of the input and observing changes in the output to infer input significance (Li et al., 2016), utilizing methods like Meaningful Perturbations (Fong & Vedaldi, 2017) and LIME (Ribeiro et al., 2016). Despite advancements, challenges such as attribution leakage (Adebayo et al., 2018), unreliability of saliency methods (Kindermans et al., 2019), and complexities in attributing outputs in large models (Ghorbani et al., 2019) persist. Surveys like Li et al. (2023) discuss current methodologies and inherent challenges, while research by Keeling & Street (2024) examines the theoretical basis for attributing confidence to LLMs, raising concerns about the reliability of experimental assessment techniques. In contrast to the discussed gradient-based attribution methods that are challenging to scale and perturbation-based approaches that need multiple passes through the model, we propose a straightforward prompting-based attribution approach that can be applied to any LLM agent, is readily scalable, and does not require multiple passes through the model.

### 2.2 Literature Review Generation

The concept of literature review generation using large language models (LLMs) is built upon the foundation laid by the Multi-XScience dataset proposed by Lu et al. (2020). This dataset paves the way for the challenging task of multi-document summarization, specifically focusing on generating the related work section of a scientific paper. As underlined by Lu et al. (2020), this approach favors abstractive models, which are well suited for the task. However, unlike the approach suggested by Lu et al. (2020), our work introduces the use of intermediate plans to improve the quality of generated literature reviews. The empirical evidence presented in our study shows that our novel strategy outperforms the vanilla zero-shot generation previously championed by the Multi-XScience dataset (Lu et al., 2020). (Note: This paragraph was entirely generated by GPT-4 following plan-based generation.[2])

Traditional methods for Natural Language Generation have typically employed a rule-based modular pipeline approach comprising of multiple stages of generation with intermediary steps of content planning (selecting content from input while also determining the structure of the output), sentence planning (planning the

---

[2]We use the plan: Please generate 5 sentences in 60 words. Cite @cite_1 at line 1, 3 and 5. We postprocess to replace delexicalized tokens with latex commands. Outputs from other models are compared later in Appendix (Tables 13 and 14).

structure of sentences) and surface realization (surfacing the text in sentence) (Reiter & Dale, 1997; Stent et al., 2004; Walker et al., 2007). Our proposed plan-based prompting technique draws a parallel between the modern methods of end-to-end neural models for joint data-to-text generation with micro or content planning (Gehrmann et al., 2018; Puduppully et al., 2019; Puduppully & Lapata, 2021) where we use plans to define the sentence structure of the generated output. While some recent works have explored planning in terms of devising actions (Yang et al., 2022; Song et al., 2023; Wang et al., 2023), prompting LLMs based on sentence plans have not been explored, to the best of our knowledge. We show two strategies of using plans 1.) The model generates the sentence plan as an intermediary step and conditions on this generated plan to output the final summary autoregressively. 2.) Humans can provide a ground-truth plan which results in an iterative setting, inherently providing controllability to the generated text where LLMs are susceptible to generating additional content.

Closely related to our work, Gao et al. (2023) generates answers for questions based on the citations from Wikipedia. Also related to our work, Pilault et al. (2020) examined LLM-based abstractive summarization of scientific papers in the arxiv dataset of Cohan et al. (2018); however, their work was limited to creating the abstract of a single document. Perhaps the most similar prior prompting-based approach to our work is known as 0-shot chain-of-thought prompting (Kojima et al., 2022; Zhou et al., 2022) where a model is prompted with 'Let's think step-by-step' (and similar prompts).

Additionally, Galactica has been developed to store, combine, and reason about scientific knowledge (Taylor et al., 2022). It outperforms existing models on various scientific tasks and sets new state-of-the-art results on downstream tasks. These findings highlight the potential of language models as a new interface for scientific research. However, the Galactica model was not developed to specifically address the problem of literature review assistance and it was not instruction fine-tuned to follow writing plans, and as such it suffered from the effect of hallucinating non-existent citations and results associated with imaginary prior work.[3] Recent works (Rodriguez et al., 2024a;b; Awadalla et al., 2024) have focused on building datasets multimodal of documents and scientific contents. However, our study focuses on exploring the zero-shot abilities of LLMs for literature review generation and proposes a novel strategy that includes generating an intermediate plan before generating the actual text. Our empirical study shows that these intermediate plans improve the quality of generated literature reviews compared to vanilla zero-shot generation. Furthermore, we ensure the validity of our experiments by using a new test corpus consisting of recent arXiv papers to avoid test set contamination. (Note: GPT-3.5 generated this paragraph with the 4th sentence added by the authors).

## 3 Retrieval of Related Work

In this section, we discuss the creation of the corpus of arXiv papers to examine different retrieval strategies for finding related works for a given paper abstract using different academic and generic web search engines, including Semantic Scholar and Google Search.

### 3.1 Dataset Construction

We create two datasets that contain papers posted on arXiv in August and December 2023, respectively, starting with 1,000 papers from each month. We use the arXiv wrapper in Python[4] to create RollingEval datasets. We then filter out papers for which we were not able to retrieve 100 relevant paper results using LLM summarized keywords. We query the Semantic Scholar API available through the Semantic Scholar Open Data Platform (Lo et al., 2020; Kinney et al., 2023) to search for the relevant papers. To get Google search results, we use SERP API,[5] specifically conditioned to leverage the "site:arxiv.org" parameter. This approach ensures the retrieval of search results are sourced solely from arXiv.org.

To combine results from multiple queries, we take the equal number of top results from each query to get a total of 100 papers. We took caution to avoid duplicate results from different queries. In case we are not able to retrieve a sufficient number of results from a query, we then take an equal number from the rest of the

---

[3]This sentence was inserted by the authors.

[4]https://pypi.org/project/arxiv/

[5]https://serpapi.com/

queries. This way we ensure that we always retrieve a candidate pool of 100 possible related work for each query paper. We pass these papers as queries to our literature review generation pipeline. We now describe our two-step retrieval mechanism and provide its pseudo-code in Algorithm 1.

## 3.2 Retrieving Candidate Papers

---

**Algorithm 1** Retrieval algorithm

---

**Require:** Input abstract $a$
1: keywords = LLMKeywords($a$); // Generate keywords from the abstract using an LLM
2: candidate_papers = SearchEngine(keywords); // Query a search engine to retrieve candidates
3: reranked_papers = LLMRerank(candidate_papers, $a$); // LLM-based reranking of candidates
4: **return** reranked_papers

---

To retrieve related work for a given paper abstract, **first,** for each query abstract in the dataset, we prompt an LLM to generate keywords that we use as queries for a search API (refer to Figure 14 in the Appendix for the detailed prompt used for this task). Importantly, we add a timestamp filter to the search API to retrieve papers published strictly before the publication date of the query paper. In addition to evaluating multiple search engines, we also experiment with generating multiple queries[6] from an LLM and various heuristics to combine the search results from each query (see Appendix D). We evaluate multiple general and academic search engines on the quality of the retrieved papers using coverage, which we define as the percentage of ground-truth papers retrieved by the search engine.

Table 1 shows the coverage for different search engines and query heuristics. We note that using multiple queries achieves the highest coverage with comparable results for Semantic Scholar search and SERP API. However, at best, we retrieve just under 7% of the ground truth papers. The low retrieval percentage of just under 7% can be attributed to several factors. First, the task of finding related work for a given paper is inherently challenging due to the diverse styles and methods authors use in literature reviews. This stylistic variability means that a one-size-fits-all approach, such as generating search keywords and using a search engine, might not capture the nuanced criteria that a human expert would apply. Additionally, our retrieval process operates in a constrained setting, generating search keywords based solely on the paper's abstract. In theory, including more of the paper—such as the introduction or methodology—could provide richer context and lead to higher retrieval rates. However, this would not align with our intended use case: supporting researchers in the early stages of drafting when only limited, unpolished material is available.

We also explore an embedding-based strategy for retrieval using SPECTER embeddings (Cohan et al., 2020). SPECTER uses a contrastive learning approach to train a SciBERT (Beltagy et al., 2019) model based on a triplet loss to discriminate between related versus unrelated papers. SciBERT is a BERT-like transformer

---

[6]In our experiments, we generate three queries for each abstract.

| Search type | RollingEval-Aug (%) | RollingEval-Dec (%) |
|---|---|---|
| arXiv API (Single query) | 0.65 | 1.41 |
| SERP API - Google Search (Single query) | 1.23 | 4.34 |
| Semantic Scholar API (Single query) | **3.93** | **4.76** |
| arXiv API (Multiple queries) | 2.75 | 1.92 |
| SERP API - Google Search (Multiple queries) | **6.80** | 5.04 |
| Semantic Scholar API (Multiple queries) | 6.07 | **5.09** |
| SPECTER2 | 8.30 | 6.80 |
| Semantic Scholar API (Multiple queries) + SPECTER2 | **9.80** | **8.20** |

Table 1: We created two datasets to measure the efficacy of search using LLM-generated keywords - RollingEval-Aug and RollingEval-Dec. We evaluate the % of the ground truth references covered in the top 100 search list using LLM-based keyword search and different academic search engines. We note that using multiple queries gives us an edge over using a single query, and we obtain a similar coverage for Semantic Scholar and the SERP API-based Google Search.

model (Devlin et al., 2018) trained on scientific text. More recently, Singh et al. (2022) further extended the contrastive learning approach of SPECTER to better deal with multiple tasks of relevance to scientific papers, including citation prediction, leading to SPECTER2. Since SPECTER2 provides a large knowledge base of 150M document embeddings of scientific articles, we explore embedding-based retrieval of candidate papers, where we obtain the top-$k$ papers based on the cosine similarity between the query abstract and the corpus of candidate papers in the SPECTER2 dataset.

### 3.3 Re-ranking Candidate Papers

**Next,** given a list of retrieved papers, we explore re-ranking the list using an LLM. The retrieved abstracts and the original query abstract are used as input to an LLM Re-ranker, which provides a listwise ranking of the candidate papers based on the relevance to the query abstract. We explore different strategies for reranking the candidates, detailed as follows: **a) Instructional permutation generation**: we use the approach by Sun et al. (2023), which prompts the model to directly generate a permutation of the different candidate papers, thus producing an ordered list of preferences against providing intermediate scores; **b) SPECTER2 embeddings**: We use SPECTER2 embeddings as an alternative to prompting-based strategies for reranking, where we rank the candidate papers based on their cosine distances to the SPECTER2 embedding of the query abstract (see Appendix D for more details on SPECTER2 implementation); and **c) Debate Ranking with Attribution (Ours)**: our prompting-based approach that builds on the work of Rahaman et al. (2024), where we pass each candidate paper's abstract along with the query abstract and instruct the LLM to (1) generate arguments for and against including the candidate paper and (2) output a final probability of including the candidate based on the arguments. Crucially, we add an attribution step to this ranking module, where we instruct the LLM to extract verbatim sentences from the candidate abstract that support the arguments, and we re-prompt the LLM if the extracted sentences are not present in the candidate abstract.

### 3.4 Retrieval and Re-ranking Experiments

We use an ensemble of search engines to retrieve candidates based on an abstract. We now describe the search engines used in our approach. Based on Table 1, we select the Semantic Scholar (S2) API as the search engine to retrieve search results using LLM-generated keywords. While the SERP API provides access to a broader set of search results, it is expensive, making it less practical for large-scale retrieval. Additionally, the S2 API is specifically designed for academic literature, offering strong performance comparable to the SERP API. As discussed in Section 3.2, we also explore SPECTER2 embeddings for retrieval and compare it with the different prompting-based strategies.

We experiment with different combinations of search engine and the retriever method, and present our results in Figure 2. We present precision and normalized recall at different values of top-$k$ recommendations, where we calculate normalized recall as the proportion of the number of ground truth papers retrieved (instead of all ground truth papers). Formally, normalized recall and precision follow the definitions:

$$\text{Normalized Recall@k} = \frac{|\text{Retrieved@k} \cap \text{Ground Truth}|}{|\text{Retrieved} \cap \text{Ground Truth}|}; \text{Precision@k} = \frac{|\text{Retrieved@k} \cap \text{Ground Truth}|}{k} \tag{1}$$

where "Retrieved" denotes the set of *all* candidate papers, "Retrieved@k" denotes the set of top $k$ candidate papers, and "Ground Truth" denotes the set of papers cited by the query paper. Unlike standard recall, which measures how many ground truth citations are retrieved, Normalized Recall@k measures how effectively the retrieval method prioritizes the most relevant papers in the top-$k$. By normalizing over the total relevant papers retrieved at any rank, this metric helps evaluate ranking quality independently of retrieval coverage. We include a working example in Appendix D.1 to further demonstrate the difference between normalized recall and standard recall.

From Figure 2, we note that debate ranking significantly outperforms permutation ranking, as denoted by the higher precision and normalized recall at smaller values of top-k recommendations; however, SPECTER

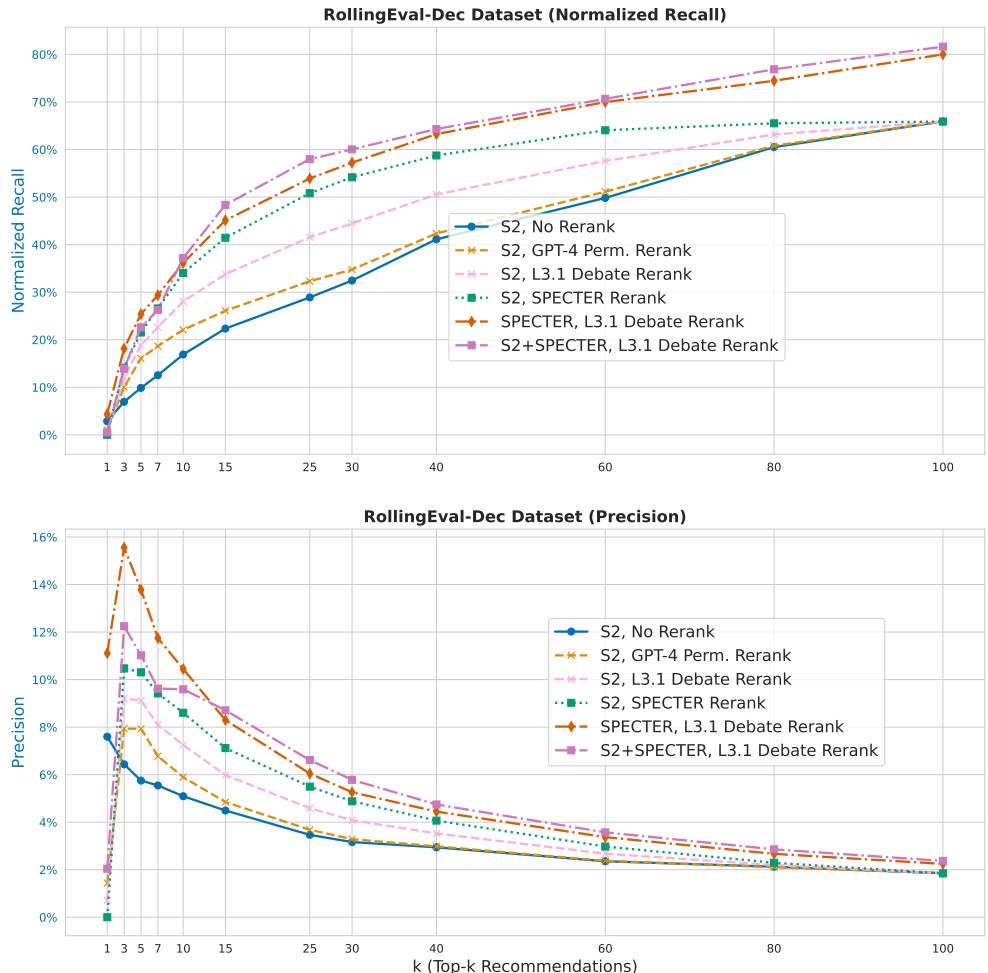

Figure 2: Effect of re-ranking strategies on the RollingEval-Dec dataset. We use the entire dataset ($n = 500$) and set $k = 100$ for these experiments. We evaluate the Precision and Normalized Recall of the re-ranked results with embedding-based ranker (SPECTER2) outperforming GPT-4 based re-ranking. We find a similar pattern for the RollingEval-Aug dataset, as shown in Appendix (Figure 7). **Note:** The first part in the legend denotes the search database for retrieval, and the second denotes the re-ranking mechanism.

embeddings outperform both prompting-based strategies. Next, the higher precision and normalized recall values for SPECTER and S2+SPECTER settings suggest that SPECTER is also an excellent search engine.

In Table 2, we examine the behaviour of the GPT-4 reranking approach in more detail. Using GPT-4 as a reranker in the manner discussed above produces an incomplete list 41% of the time. In 3% cases, GPT-4 produces a list with repeated values and, in some rare cases, with garbage values or numbers (e.g. 2020). We conclude that this strategy of using GPT-4 for reranking is brittle.

### 3.5 Positive Effect of Attribution Verification

We conduct an ablation study on the first 100 papers of the larger set of 500, and focus on the top $k = 40$ papers, which is representative of the typical number of papers cited in the Machine Learning community. In Figure 3 we show the result of removing the verification step in our debate re-ranking strategy, i.e. in this ablation, we do not check if the sentences extracted by the LLM are indeed present in the candidate paper abstract. We find that removing this kind of attribution verification leads to a drop in the precision and normalized recalls, especially for lower $k$ values. We perform a t-test to test the significance of the drop and

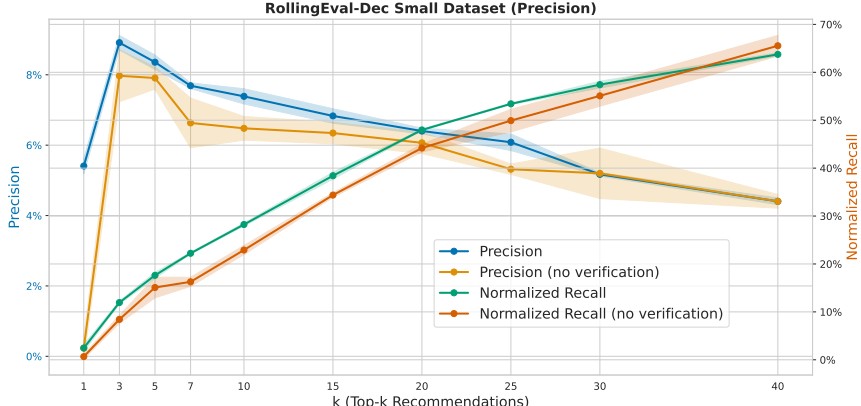

Figure 3: The effect of removing the referenced content verification step in our debate ranking strategy. We plot precision and normalized recall for two variants of the debate ranking strategy. For this ablation study, we select a smaller subset of $n = 100$ query abstracts, set $k = 40$, and repeat the experiment for three random seeds. We plot the mean and show the standard deviation as the shaded region. We find that the precision and normalized recall drop slightly upon removing the verification step. This difference is significant (as determined by the t-test,) indicating that the verification step is crucial for the success of the debate ranking strategy.

find that for both precision and normalized recall, the drop is significant, with $p$-values of $4.7 \times 10^{-4}$ and $1.9 \times 10^{-6}$ for precision and normalized recall curves, respectively. This indicates that proper attribution also allows the LLM to provide a more accurate ranking of candidates.

| Ranker Predictions | RollingEval-Aug (%) | RollingEval-Dec (%) |
|---|---|---|
| Complete Ranked list | 55.1 | 59.7 |
| Incomplete list | 41.5 | 40.2 |
| Repeated Value | 3.3 | 0.1 |

Table 2: Error modes of GPT-4 based ranking. When using GPT-4 to provide ranks, it suffers from multiple issues. While recent works like Sun et al. (2023) explore LLMs like GPT-4 for 0-shot rank predictions, we find a tendency of GPT-4 to produce an incomplete list or repeated values in the re-ranked order list with some rare cases of garbage values.

## 4 Literature Review Generation

**Plan Based Generation Approach & Model Variants.** We now focus on generating the related work section of a scientific document from a user-supplied list of papers. In a real-world scenario, this list might be obtained through traditional means, from the above-mentioned automated methods, or some combination. We evaluate several dimensions of writing quality in the generated text. Importantly, while modern LLMs can yield seemingly well-written text passages, "hallucinations" remain a problem and can be particularly egregious when LLMs are used in scientific writing (Athaluri et al., 2023). The hallucination of statements not entailed by the contents of cited papers and the hallucinations of imaginary papers that do not exist is a well-known issue of even the most powerful LLMs. We use ideas from retrieval augmented generation (RAG) techniques (Lewis et al., 2020) and instruction prompting to address the key problem of hallucinations. Our work also aims to increase the number of papers from the desired set that are indeed discussed (the coverage).

We present our general framework and problem setup in Figure 4. We use the abstract of a query paper — the one for which we generate a literature review, along with the abstracts of the set of papers to be cited (the retrieved abstracts of reference papers) to generate the related work section of the query paper. For evaluation, our approach relies on prompting LLMs in different ways and measuring the similarity of the

generated literature review text to ground truth literature reviews found within a corpus of recent scientific papers — i.e. ones not used in the training set of the underlying LLMs. We use both automated methods and human evaluations in our experiments below.

We propose to further decompose the writing task to increase passage quality, factual accuracy, and coverage. We examine different strategies for generating a *writing plan*, a line-by-line description including citations of the passage to write. These writing plans also give authors (users) control over the output passages. This is likely essential in practice to meet author preferences and possible publication constraints. These plans are defined and generated in such a way that the user can interact with and edit them if desired, or they can be used as an automatically generated intermediate (but human-understandable) representation. We now describe our proposed methods, their use in practice and their evaluation in more detail.

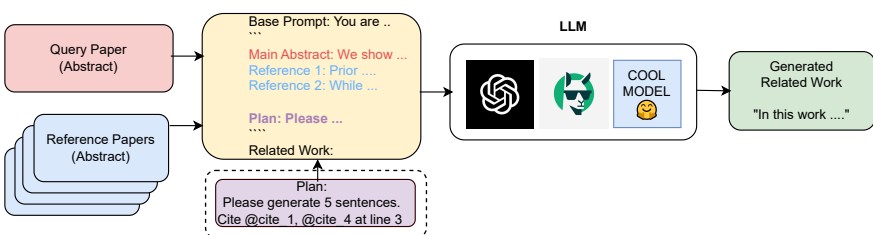

Figure 4: Pipeline of generation task where the model needs to generate the related work of the query paper conditioned on reference papers. Our method employs an optional plan — shown by the dotted purple box, either generated by the model or appended to the prompt.

In what we refer to as plan-based generation (**Plan**), a model is prompted with a known (user-provided) plan to produce X sentences in Y words and cite references on specific lines. These plans are obtained from the ground truth data and serve as a proxy for a user who desires text to be written with specific constraints. This kind of Plan strongly guides generated text to the user's desires (as observed by the final ground truth text). During evaluation, this might be considered a form of teacher-forcing at the structural level. An example of the format of these plans is provided below:

```
Please generate {num_sentences} sentences in {num_words} words.  Cite {cite_x} at line {line_x}.  Cite {cite_y} at line
{line_y}.
```

**Prompted plan.** The previous method replicates the scenario of a detailed user-provided plan. However, one can also generate such plans automatically from an LLM. The model is prompted first to generate a plan of sentences and citations, which it would then condition upon to generate the final related work text. When used as an interactive tool, we envision the user might start with a suggested plan, see the corresponding generated full literature review text, and then iteratively edit the plan and regenerate the result. See our Appendix (Figures 9, 10 and 11) for the differences in the prompts used in our experiments. We also experiment with two other strategies in which researchers could prompt the model:

**Per cite.** We first use a two-stage strategy to generate content relevant to each cited paper. In the first stage, the model is prompted to generate related works in 1-2 lines for each individual reference citation. All the outputs for different citations are combined together to form the generated related work. In the second stage, the LLM summarizes and paraphrases the output of the first stage.

**Sentence by sentence** Based on the Ground truth (GT) related work and the citation on each line, we prompt the model to generate one sentence conditioned on the abstract, the reference cited in that line, and the generated draft so far. In the absence of a citation or at the start, the model is prompted only with the abstract and draft of the generated work till now.

# 5    Generation Experiments.

For the following studies on generating related work, we introduce an additional corpus. We extend the Multi-XScience corpus (Lu et al., 2020) to include the full text of research papers. We also reuse the RollingEval-Aug introduced in Section 3. We use HuggingFace Transformers (Wolf et al., 2019) and PyTorch (Paszke et al., 2017) for our experiments[7] and calculate ROUGE scores (Lin, 2004) using the Huggingface's `evaluate` library. Details on the dataset and the implementation are in Appendix B and D, respectively. Similar to Lu et al. (2020), we extract the ground truth cited references as the relevant papers and evaluate only the generated outputs from different systems. Since ROUGE score only measures token-level similarity and does not account for semantic meaning, we also report BERTScore and Llama-3-Eval in Table 4. We use BERTScore, an embedding-based evaluation metric, to account for the semantic meaning of two texts during evaluation as well. On the other hand, we use Llama-3-Eval, an open-source variant of the widely used G-Eval metric (Liu et al., 2023b), as G-Eval has been shown to correlate better with human preferences.

## 5.1    Generation Baselines

**Extractive baselines** As in Lu et al. (2020), we report the performance of LexRank (Erkan & Radev, 2004) and TextRank (Mihalcea & Tarau, 2004). We also create a simple one-line extractive baseline which extracts the first line of the abstract and combines all the citations to form the output.

**Abstractive finetuned baselines** We use the model outputs of Hiersum (Liu & Lapata, 2019) and Pointer-Generator (See et al., 2017) from Lu et al. (2020) for abstractive finetuned baselines. We also reproduce the finetuned PRIMER (Xiao et al., 2021) model (considered to be the SOTA).

**Abstractive zero-shot baselines** We use the zero-shot single-document abstractive summarizers FlanT5 (Chung et al., 2022) and LongT5 (Guo et al., 2022) based on the T5 architecture (Raffel et al., 2020). Since Galactica (Taylor et al., 2022) is trained on documents from a similar domain, we include it along with Falcon-180B (Almazrouei et al., 2023).

**Open and closed source models** We use different chat versions (7B, 13B, 70B) of Llama 2-Chat[8] (Touvron et al., 2023) as zero-shot open-source LLM baselines. For closed-source models, we evaluate zero-shot both GPT-3.5-turbo (Brown et al., 2020) and GPT-4 (OpenAI, 2023). Since they perform best in our initial evaluations, we use the closed-source models in combination with the different generation strategies (Per-cite, Sentence by sentence, plan-based, and learned plan) from Section 4.

# 6    Results and Observations

From Table 3, we first note that unsupervised extractive models provide a strong baseline compared to abstractive 0-shot single document summarization baselines. Fine-tuning these abstractive models on Multi-XScience (released initially with the benchmark) improves performance at least to the level of extractive models. We reproduce the PRIMER model using their open-source code but find lower-than-reported results. As such, we consider the Pointer-Generator method to be the current state-of-the-art (SOTA).

Single-document summarizers (LongT5, Flan T5) perform poorly in the zero-shot settings with limited ability to cite references. We are limited in the prompt we can provide (because of the training prompts) and resort to "Summarize the text and cite sources." Galactica's performance is encouraging compared to other models in the same group, but inspecting its output reveals that it generates the whole introduction of the paper instead of the related work. The model is very sensitive to the prompts used (mostly as suffixes) and struggles to follow instructions. Falcon 180-B, on the other hand, tends to hallucinate user turns and considers this task as multiple turns of user-system exchange, even though we prompted to generate relevant outputs.

All recent versions (7B, 13B, 70B) of zero-shot Llama 2 models underperform the supervised Pointer-Generator baseline (except for 70B on ROUGE2) and their GPT counterparts. All Llama 2 models tend to produce output in bullet points and also provide references. We find that closed-sourced models like GPT-3.5-turbo and

---

[7]Code can be accessed at https://github.com/LitLLM/litllms-for-literature-review-tmlr

[8]We refer to Llama 2-Chat models as Llama 2 and GPT-3.5-turbo as GPT-3.5 for brevity.

| Model Class | Model | ROUGE1 ↑ | ROUGE2 ↑ | ROUGEL ↑ |
|---|---|---|---|---|
| Extractive | One line baseline | 26.869 | 4.469 | 14.386 |
| | LexRank | 30.916 | 5.966 | 15.916 |
| | TextRank | 31.439 | 5.817 | 16.398 |
| Abstractive Finetuned | Hiersum | 29.861 | 5.029 | 16.429 |
| | Pointer-Generator | 33.947 | 6.754 | 18.203 |
| | PRIMER | 26.926 | 5.024 | 14.131 |
| Abstractive 0-shot | Long T5 | 19.515 | 3.361 | 12.201 |
| | Flan T5 | 21.959 | 3.992 | 12.778 |
| | Galactica-1.3B | 18.461 | 4.562 | 9.894 |
| | Falcon-180B | 22.876 | 2.818 | 12.087 |
| Open-source 0-shot | Llama 2-Chat 7B (No plan) | 24.636 | 5.189 | 13.133 |
| | Llama 2-Chat 13B (No plan) | 26.719 | 5.958 | 13.635 |
| | Llama 2-Chat 70B (No plan) | 28.866 | 6.919 | 14.407 |
| | LLama-3.1-70B (No Plan) | 33.289 | 8.050 | 15.898 |
| Closed-source 2-stage | GPT-3.5-turbo (Per cite) 1st stage | 26.483 | 6.311 | 13.718 |
| | GPT-3.5-turbo (Per cite) 2nd stage | 24.359 | 5.594 | 12.859 |
| | GPT-3.5-turbo (Sentence by sentence) | 31.654 | 6.442 | 15.577 |
| Closed-source 0-shot | GPT-3.5-turbo (No plan) | 29.696 | 7.325 | 14.562 |
| | GPT-4 (No plan) | 33.213 | 7.609 | 15.798 |
| Plan | Llama 2-Chat 70B (Prompted plan) | 30.389 | 7.221 | 14.911 |
| | GPT-3.5-turbo (Prompted plan) | 32.187 | 7.788 | 15.398 |
| | GPT-4 (Prompted plan) | 34.819 | 7.892 | 16.634 |
| | Llama 2-Chat 70B (Plan) | 34.654 | 8.371 | 17.089 |
| | GPT-3.5-turbo (Plan) | 35.042 | 8.423 | 17.136 |
| | GPT-4 (Plan) | 37.198 | 8.859 | 18.772 |
| | Llama-3.1-70B (Plan) | 35.575 | 9.406 | 18.772 |

Table 3: Zero-shot results for different models on the Multi-XScience dataset.

| Model | ROUGE1 ↑ | ROUGE2 ↑ | ROUGEL↑ | BERTScore↑ | Llama-3-Eval↑ |
|---|---|---|---|---|---|
| CodeLlama 34B-Instruct | 22.608 | 5.032 | 12.553 | 82.418 | 66.898 |
| CodeLlama 34B-Instruct (Plan) | 27.369 | 5.829 | 14.705 | 83.386 | 67.362 |
| Llama 2-Chat 7B | 23.276 | 5.104 | 12.583 | 82.841 | 68.689 |
| Llama 2-Chat 13B | 23.998 | 5.472 | 12.923 | 82.855 | 69.237 |
| Llama 2-Chat 70B | 23.769 | 5.619 | 12.745 | 82.943 | 70.980 |
| GPT-3.5-turbo (0-shot) | 25.112 | 6.118 | 13.171 | 83.352 | 72.434 |
| GPT-4 (0-shot) | 29.289 | 6.479 | 15.048 | 84.208 | 72.951 |
| Llama 2-Chat 70B (Plan) | 30.919 | 7.079 | 15.991 | 84.392 | 71.354 |
| GPT-3.5-turbo (Plan) | 30.192 | 7.028 | 15.551 | 84.203 | 72.729 |
| GPT-4 (Plan) | 33.044 | 7.352 | 17.624 | 85.151 | 75.240 |

Table 4: Zero-shot results on the proposed RollingEval-Aug dataset.

GPT-4 achieve SOTA in the zero-shot setting. However, the proposed sentence-by-sentence and per-citation strategies deteriorate the performance of GPT models which tends to cover all related concepts hierarchically.[9]

Our teacher-forced plan-based framework improves the scores over the 0-shot baseline for both closed-sourced (GPT-3.5 and GPT-4) and open-sourced LLMs, with Llama 2 70B achieving similar scores as GPT-3.5 on both the original Multi-XScience and the new RollingEval-Aug dataset (in Table 4). In Table 4, we also notice that BERTScore and Llama-3-Eval exhibit the same trends as ROUGE scores except in the case where GPT-3.5-turbo (Plan) obtains a higher Llama-3-Eval score than Llama 2-Chat 70B (Plan). These values also showcase the weaker discerning power of BERTScore compared to Llama-3-Eval as all the models achieve a high BERTScore between 82-85%. Llama 2 70B gets more uplift with the plan compared to GPT models where manual inspection reveals fewer hallucinations in the outputs (see qualitative results in Table 13 in Appendix using our abstract). In Table 5, we evaluate the controllability of LLM-based generation using sentence plans and find that GPT-4 tends to follow the plan more closely. It follows the exact plan 60% of the time, often producing fewer sentences than provided. Llama 2 70B comes in second place in following the plan instructions and GPT-3.5 struggles to follow the plan precisely. We also experiment with a learned plan strategy where the model first generates a plan and then autoregressively generates the output. Though it

---

[9]We validated these strategies only on GPT-3.5 due to the high incurred cost with GPT-4 models.

| Model | Multi-XScience | | | RollingEval-Aug | | |
|---|---|---|---|---|---|---|
| | % ↑ | Mean ↓ | Max ↓ | % ↑ | Mean ↓ | Max ↓ |
| GPT-3.5-turbo (Plan) | 4.73 | 3.65 | 17 | 3 | 4.7 | 16 |
| Llama 2-Chat 70B (Plan) | 19.04 | 2.66 | 22 | 17.4 | 2.72 | 18 |
| GPT-4 (Plan) | 60.7 | −0.01 | 8 | 70.6 | 0.16 | 5 |

Table 5: We show % of responses with the same number of lines as the plan for both datasets. Here we also show the mean and max difference in lines generated by the model vs. the original plan. -ive implies that a model generated fewer lines than the plan. We find GPT-4 to follow the plan more closely compared to Llama 2 and GPT-3.5.

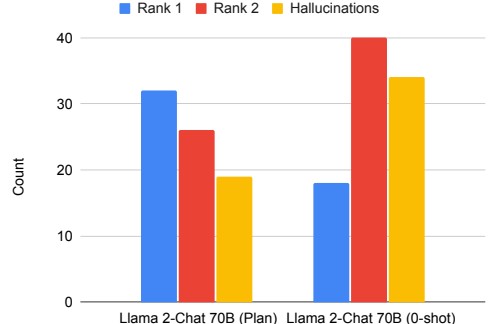 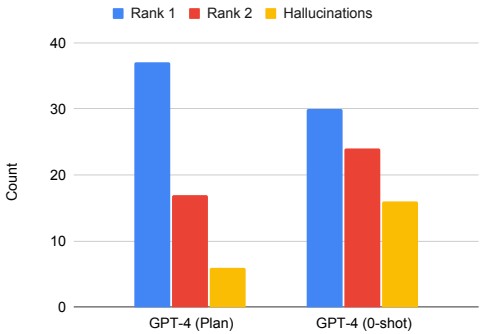

Figure 5: Human evaluation study where annotators ranked the generations of 0-shot models with their sentence-plan-based counterparts. On the Y-axis, we show counts from an overall sample size of 58 annotations for Llama 2-Chat and 54 for GPT-4 (where ranking ties are allowed). We see a reduction of 58.6% cases of hallucinations to 32.7% for Llama 2-Chat and 29.6% to 11.6% for GPT-4 using plan-based prompting.

improves the results over 0-shot baseline, it does not outperform the teacher-forced plan generation in terms of automatic metrics. There is a considerable drop in performance on RollingEval-Aug dataset compared to the original Multi-XScience in terms of ROUGE1/2. It gives more credibility to the hypothesis that the Multi-XScience test set is in the training data of these LLMs and/or that there is a shift in the distribution of these more recent papers. Nevertheless, we find similar scores and ranking patterns between models as for Multi-XScience. We provide other experiments related to fine-tuning, longer context, and CodeLLM in Appendix C, and we provide cost estimates for different methods in Appendix D.5.

**Coverage and human evaluation:** We evaluate coverage as the percentage of model outputs with the same number of citations as ground truth (identified using regex on the delexicalized citation in the generated related work). Table 6 shows the efficacy of plan-based approaches. All plan models provide more coverage than their counterparts, with GPT-4 achieving 98% in covering all the citations. The largest uplift is for (vanilla) 0-shot Llama 2 70B. Using a plan raises its coverage from 59% to 82%. Similar to results in Table 5, we find that the coverage of GPT-3.5 does not improve much.

| Model | Coverage ↑ | Avg. words |
|---|---|---|
| Llama 2-Chat 70B (0-shot) | 59.31% | 284.65 |
| Llama 2-Chat 70B (Plan) | 82.62% | 191.45 |
| GPT-3.5-turbo (0-shot) | 63.11% | 293.69 |
| GPT-3.5-turbo (Plan) | 68.03% | 202.81 |
| GPT-4 (0-shot) | 91.34% | 215.15 |
| GPT-4 (Plan) | 98.52% | 125.10 |

Table 6: Coverage (in %) on the Multi-XScience dataset defines the number of citations covered in the generated response.

We also run a human evaluation study using 6 expert annotators. They rank model generated outputs for 160 papers with 3 citations[10] where we show the abstract of the query paper, cited references, and the model outputs of 0-shot vs plan counterparts side-by-side. We randomly selected 80 examples each for GPT-4 and Llama 2-Chat comparisons. Experts had the flexibility to choose rank 1 for one or both the models, otherwise they could select rank 2 for each. We also ask the annotators to identify hallucinations, i.e. which model generated content not from the abstracts. The interface is described further and shown in the Appendix (Figure 8). Out of 160, we find agreement among the annotators for 112 examples (54 for GPT4 and 58 for Llama 2-Chat). The results in Figure 5 show that humans rank generated response to be significantly better[11] for Llama 2-Chat Plan based model where it was ranked at the top 32 times compared to 18 times for 0-shot. We can also observe a similar trend for GPT-4, where annotators rank GPT-4 Plan-based output 37 times at Rank 1 compared to 30 for GPT-4 0-shot. In terms of hallucinations, we find significant reductions in the hallucination for plan-based Llama 2-Chat models from 34 to 19 instances, where the 0-shot model often provided a made-up citation (XYZ et al.), possibly from background knowledge. Only 6 cases of hallucination were found for GPT-4 plan compared to 16 instances for 0-shot vanilla GPT-4 model.

# 7 Conclusions & Answering: Are We There Yet?

This work discusses, establishes and evaluates a pipeline to help people write literature reviews. We first identified some challenges of evaluating such systems when LLMs are constantly updated based on training on new data, which may contain recent papers found online. To address these issues, we propose and implement a rolling evaluation procedure that focuses on recent arXiv papers, and we collect several evaluation datasets in this manner.

Our experiments show that LLMs have significant potential for writing literature reviews, especially when the task is decomposed into these smaller and simpler sub-tasks that are within reach of LLMs, namely through the use of LLM-generated keyword search and embedding-based search for relevant prior work. Notably, our experiments indicate that both debate prompting and debate arguments that use attribution based on citing extracted content from source material improve LLM re-ranking results. The most powerful LLMs evaluated in our studies exhibit extremely promising paper re-ranking abilities as well as promising literature review generation results. Importantly, LLM hallucinations can be substantially reduced using our proposed plan-based prompting and retrieval augmented generation techniques.

Our evaluation also reveals clear challenges: 1) retrieving all relevant papers consistent with a given human-generated literature review will require new querying strategies; 2) hallucinations can be significantly reduced using plan-based prompting, but our approach does not completely eliminate hallucinations.

So, *are we there yet? Not quite—but we are getting closer.* The landscape of AI-assisted research exploded in 2025, with tools like DeepResearch (OpenAI, 2025), AI Co-Scientist (Gottweis et al., 2025), and ScholarQA (AllenAI, 2025) demonstrating remarkable improvements in literature review generation, citation accuracy, and retrieval strategies. Moreover, to accompany our scientific work here, and our older work on this theme (Agarwal et al., 2024), we have build a full working demo based on our proposed retrieval and generation pipeline (see Figure 16 in the Appendix for a screenshot), which we will release to the community. We hope that authors can use this demonstration system to better understand how these techniques—and future alternatives—can be most helpful for assistance in literature review generation.

**Limitations and Future Work.** Because of the low coverage for retrieval, we evaluate different components independently. During generation, this strategy assumes that we already have filtered relevant papers corresponding to the main paper. In the future, we would like to improve the search for relevant work using embedding-based models to get better coverage and, thus, the ability to evaluate the system end-to-end. Our retrieval component currently suffers from surface-level information about the papers, while in practice, authors frame search keywords based on information (such as underlying datasets) that might not be present in the abstract. Another issue stems from the retrieval evaluation setup based on coverage related to the

---

[10]This was done to reduce cognitive load on annotators to read abstracts of 4+ papers and 2 model outputs. Annotators have research experience in machine learning at the Masters or Doctorate level.

[11]We used McNemar test (Lachenbruch, 2014) to measure statistical significance.

ground truth papers, where the authors might have different biases. We also acknowledge that including more of the paper, like introduction and methodology, might help improve the set of initial candidate papers; however, using additional sections could inadvertently allow the model to detect explicit citations or references, essentially "cheating" by using these as hints to retrieve specific papers. By focusing on the abstract alone, we maintain a more controlled setup that reflects a realistic, early-stage research scenario. It is important to highlight that while we operate in this abstract-only setup, our pipeline is designed to be flexible and interactive. As the paper matures and more content becomes available, researchers can provide additional context to the pipeline to improve retrieval accuracy. This adaptability ensures that our approach remains relevant and effective throughout different stages of the research process, allowing for incremental refinement of related work as drafts evolve.

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

## Appendix

## A    Ethics Statement

The rapid advancements in LLMs and NLP technologies for scientific writing have led to the emergence of increasingly powerful systems such as DeepResearch, AI Co-Scientist, and ScholarQA. These tools extend beyond earlier systems like Explainpaper and Writefull[12], which assist in paper comprehension and abstract generation, and Scite[13], which helps with citation discovery. As AI-powered tools become more deeply integrated into the scientific workflow, ethical considerations around their use continue to evolve. Many conferences, such as ICLR, have begun collecting statistics on authors' usage of LLMs for literature review generation and paraphrasing, and have issued guidelines on responsible usage.[14] While writing assistant technology could have great promise as an aide to scientists, we think their use should be disclosed to the reader. As such assistants become more powerful, they might be abused in certain contexts, for example, where students are supposed to create a literature review as a part of their learning process. The use of such tools might also be problematic as authors of scientific work should read the articles that they cite, and heavy reliance on such tools could lead to short-term gains at the cost of a deeper understanding of a subject over the longer term. Any commercially deployed systems authors use should also contain appropriate mechanisms to detect if words have been copied exactly from the source material and provide that content in a quoted style. Additionally, as newer tools like DeepResearch, AI Co-Scientist, and ScholarQA continue to improve, it is crucial to assess their long-term impact on scientific research. The use of these tools should complement, rather than replace, human expertise in literature analysis. Finally, the rolling evaluations we present here do not involve training LLMs on arXiv papers. This mitigates concerns regarding the copyright status of arXiv papers and their use for LLM training.

## B    New Datasets

While there are datasets available for different tasks in academic literature (see Table 7), we use the Multi-XScience dataset (Lu et al., 2020) for our experiments. Recent work (Chen et al., 2021b; Funkquist et al., 2022) also focuses on related work generation and provides a similar dataset. As part of this work, we release two corpora: 1. We extend the Multi-XScience corpus to include the full text of research papers, and 2. We create a new test corpus, RollingEval-Aug, consisting of recent (August 2023) arXiv papers (with full content).

| Dataset | Task |
| --- | --- |
| BigSurvey-MDS (Liu et al., 2023a) | Survey Introduction |
| HiCaD (Zhu et al., 2023) | Survey Catalogue |
| SciXGen (Chen et al., 2021a) | Context-aware text generation |
| CORWA (Li et al., 2022) | Citation Span Generation |
| TLDR (Cachola et al., 2020) | TLDR generation |
| Multi-XScience Lu et al. (2020) | Related Work Generation |

Table 7: Different tasks for academic literature

**Multi-XScience full text** We create these datasets based on the latest release (2023-09-12) of the S2ORC corpus[15] (Lo et al., 2020) available at the Semantic Scholar Open Data Platform (Kinney et al., 2023). The S2 Platform provides access to multiple datasets, including paper metadata, authors, S2AG (Semantic Scholar Academic Graph), paper embeddings, etc. While the 'Papers' dataset consists of 200M+ metadata records, S2ORC consists of 11+M full-text publicly available records with annotations chunked into 30 files

---

[12]https://www.explainpaper.com/, https://x.writefull.com/

[13]https://scite.ai/

[14]ICLR'24 Large Language Models guidelines https://iclr.cc/Conferences/2024/CallForPapers

[15]Dataset available at http://api.semanticscholar.org/datasets/v1/

(~215G compressed json) where research documents are linked with arXiv and Microsoft Academic Graph (MAG) (Sinha et al., 2015) IDs, when available. This corpus provides full text of the research papers (parsed using a complex pipeline consisting of multiple LaTeX and PDF parsers such as GROBID (Lopez, 2023) and in-house parsers.[16]). The full text is also aligned with annotation spans (character level on the full text), which identify sections, paragraphs, and other useful information. It also includes spans for citation mentions and the matching semantic corpus-based ID for bibliographical entries, making it easier to align with references compared to other academic datasets such as LoRaLay (Nguyen et al., 2023), UnarXive (Saier & Färber, 2020; Saier et al., 2023), etc. or relying on citation graphs like OpenAlex (Priem et al., 2022), next-generation PDF parsers (Blecher et al., 2023) or other HTML webpages.[17] For the Multi-XScience, we obtain the full text of papers for 85% of records from the S2ORC data using the span annotations from the corpus aligned with citation information.

**RollingEval datasets** Llama 2 was publicly released on 18th July 2023 and GPT-4 on 14 March 2023. Both provide limited information about their training corpus, and academic texts in the Multi-XScience may or may not have been part of their training data. To avoid overlap with the training data of these LLMs, we process a new dataset using papers posted after their release date. To do so, we first filter the papers published in August 2023 from S2ORC that contain an arXiv ID, resulting in ~15k papers. S2ORC does not provide the publication date of the papers directly, so we use regex '2308' on the arXiv ID to extract papers posted in 08'23. We then use section annotations to get the section names and match using synonyms ('Related Work, Literature Review, Background') to extract section spans. We take the rest of the text as conditioning context except the related work section which results in ~4.7k documents. Using the citation annotations, we extract the full text of cited papers from the S2ORC corpus again using corpus ID. Similar to Multi-XScience, we use paragraph annotations to create a dataset for the latest papers (~6.2k rows). We create a subset of 1,000 examples (RollingEval-Aug) where we have the content of all the cited papers. The average length of a related work summary is 95 words, while the average length of abstracts is 195. On average, we have 2 citations per example, which makes the dataset comparable to the original Multi-XScience dataset.

## C  Other Generation Experiments

**Llama 2 fine-tuning** In parallel, we also fine-tune Llama 2 models on the train set with the original shorter context, but they are very sensitive to hyperparameter configuration. When we instruct-finetune Llama 2 7B, it initially produces code. We find a slight improvement when fine-tuning the Llama 2 7B model for 30k steps with an LR of 5e-6 over 0-shot model (see Table 8), but it quickly overfits as we increase the LR or the number of steps. We leave hyperparameter optimization, fine-tuning larger models with RoPE scaling and plan-based generation for future work.

| Model | ROUGE1 ↑ | ROUGE2 ↑ | ROUGEL ↑ |
|---|---|---|---|
| Llama 2-Chat 7B - 0-shot | 26.719 | 5.958 | 13.635 |
| Llama 2-Chat 7B - 10k steps (LR 5e-6) | 24.789 | 5.986 | 12.708 |
| Llama 2-Chat 7B - 30k steps (LR 5e-6) | 27.795 | **6.601** | 14.409 |
| Llama 2-Chat 7B - 60k steps (LR 1e-5) | 22.555 | 5.511 | 11.749 |

Table 8: Results after fine-tuning Llama 2-Chat 7B on Multi-XScience dataset

**Longer context** While Llama 2 can ingest 4096 tokens, recent studies have found that it uses 19% more tokens (Kadous, 2023) than GPT-3.5 or GPT-4 (2048 and 4096 tokens respectively), implying that the effective number of words in Llama 2 is lower than GPT-4 and only a bit higher than GPT-3.5. We experiment with the popular RoPE scaling (Su et al., 2021) in 0-shot Llama models to increase the context length (4k–6k). This permits using the full text of the papers instead of just their abstracts. Results in Table 9 show that directly using RoPE scaling on 0-shot models produces gibberish results. Instead, one needs

---

[16]https://github.com/allenai/papermage

[17]https://ar5iv.labs.arxiv.org/ and https://www.arxiv-vanity.com/

to fine-tune the model with the longer context. In fact, a plan-based-longer-context CodeLlama (initialized from Llama 2 and trained with a 16k token context through RoPE scaling) improves on ROUGE1/L, but achieves comparable results as a shorter-context plan-based CodeLlama on ROUGE2. For reporting results with longer context Llama 2 using RoPE scaling (Su et al., 2021), we use HuggingFace Text Generation Inference.[18]

| Model | ROUGE1 ↑ | ROUGE2 ↑ | ROUGEL ↑ |
|---|---|---|---|
| Llama 2-Chat 7B (4000 words) | 17.844 | 1.835 | 10.149 |
| Llama 2-Chat 7B (5000 words) | 17.254 | 1.736 | 9.986 |
| Llama 2-Chat 7B (6000 words) | 17.179 | 1.647 | 9.897 |
| Llama 2-Chat 13B (4000 words) | 20.071 | 3.516 | 10.916 |
| Llama 2-Chat 13B (5000 words) | 20.722 | 3.714 | 11.13 |
| Llama 2-Chat 13B (6000 words) | 17.179 | 1.647 | 9.897 |
| Llama 2-Chat 70 (4000 words) | 19.916 | 2.741 | 10.456 |
| Llama 2-Chat 70B (5000 words) | 19.675 | 2.605 | 10.48 |
| Llama 2-Chat 70B (6000 words) | 20.437 | 2.976 | 10.756 |
| CodeLlama 34B-Instruct (4000 words) | 27.425 | 5.815 | 14.744 |

Table 9: Zero-shot results using RoPE scaling for larger context on RollingEval-Aug dataset. Here we report the max number of words used for truncation instead of the tokens.

**Code LLMs** We evaluate the performance of code-generating LLMs to write related-work sections requiring more formal and structured language. Since Code LLMs are pre-trained on text they might offer the best of both worlds. However, we observe that for our task, the models produce bibtex and Python code with relevant comments as part of the generated outputs. As shown in Table 10, CodeLlama (34B Instruct) is good at following instructions and at generating natural language (ROUGE2 of 5.8 and 5.02 on Multi-XScience and RollingEval-Aug dataset). With a plan, CodeLlama even surpasses vanilla 0-shot Llama 2 70B (Table 4).

| Model | ROUGE1 ↑ | ROUGE2 ↑ | ROUGEL ↑ |
|---|---|---|---|
| StarCoder | 12.485 | 1.104 | 6.532 |
| Lemur-70B | 15.172 | 2.136 | 7.411 |
| CodeLlama 34B-Instruct | 25.482 | 5.814 | 13.573 |

Table 10: 0-shot results using code-based models on Multi-XScience dataset. CodeLlama performs reasonably well in generating natural language compared to the other code-based counterparts.

# D More implementation details

## D.1 Normalized Recall v/s Standard Recall: A Worked-out Example

Consider a query paper with the following statistics:

$$|\text{Ground Truth}| = n_{\text{gt}} = 84$$
$$|\text{Retrieved}| = 100$$
$$\text{Relevant Retrieved papers} = |\text{Retrieved} \cap \text{Ground Truth}| = c = 10$$
$$\text{Relevant papers in top-40} = n_{\text{rel}} = 4$$

Using these values, we compute the metrics at $k = 40$:

$$\text{Precision@40} = \frac{n_{\text{rel}}}{40} = \frac{4}{40} = 0.010; \quad \text{Normalized Recall@40} = \frac{n_{\text{rel}}}{c} = \frac{1}{10} = 0.100; \quad \text{Recall} = \frac{n_{rel}}{n_{gt}} = \frac{4}{84} = 0.048$$

---

[18]https://github.com/huggingface/text-generation-inference

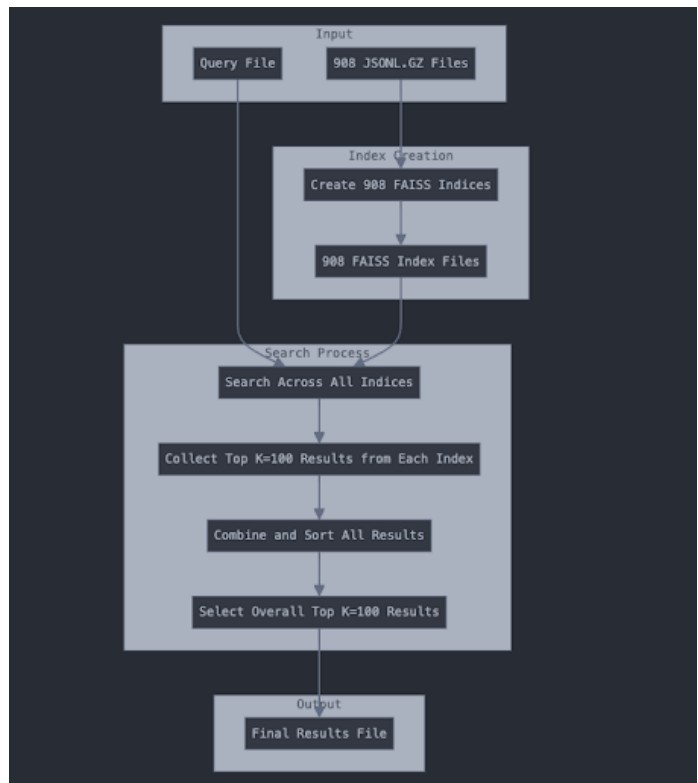

Figure 6: Pipeline for creating FAISS indexes for 150M SPECTER2 embeddings.

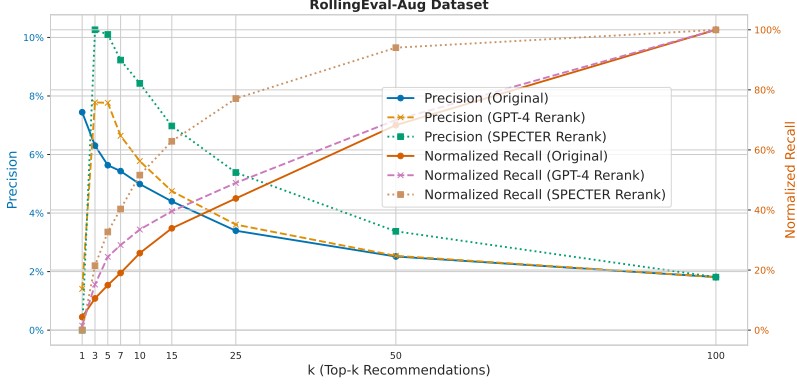

Figure 7: Effect of re-ranking strategies on the RollingEval-Aug dataset. We evaluate the Precision and Normalized Recall of the re-ranked results contrasting LLM-based based re-ranking with embedding-based ranker. We find a similar pattern as the RollingEval-Aug dataset.

This example illustrates how Normalized Recall@k differs from standard recall. Instead of being limited by the total number of ground truth citations, it evaluates how well the method ranks the retrievable relevant papers. In this case, despite a low precision, the normalized recall is relatively high, indicating that the method effectively ranks the relevant papers it does retrieve.

## D.2 Generation Implementation

We use HuggingFace Transformers and PyTorch (Paszke et al., 2017) for our experiments.[19] We calculate ROUGE scores (Lin, 2004) using the Huggingface (Wolf et al., 2019) evaluate library[20]. To split sentences, we use 'en_core_web_sm' model from SpaCy[21]. Additionally, we use Anyscale endpoints[22] to generate 0-shot Llama 2 results and OpenAI API[23] to generate results for GPT-3.5-turbo and GPT-4.

## D.3 Demo implementation

We build our system using the ReactJS framework, which provides a nice interface to build system demos quickly and efficiently. More details about the demo implementation can be found in our system paper Agarwal et al. (2024) and the project page.

We query the Semantic Scholar API available to search for the relevant papers. Specifically, we use the Academic Graph[24] and Recommendations[25] API endpoint. We use OpenAI API to generate results for LLM using GPT-3.5-turbo and GPT-4 models. At the same time, our modular pipeline allows using any LLM (proprietary or open-sourced) for different components. We also allow the end-user to sort the retrieved papers by relevance (default S2 results), citation count, or year. More details about the demo system can be found in our system paper.

## D.4 SPECTER Implementation

We build an index of 150M SPECTER2 embeddings that we can use as an alternative to both a search engine and a prompting-based ranking module. Figure 6 shows our pipeline for creating the index. Specifically, the SPECTER2 database comes with 908 json.gz files containing compressed embeddings. For each json.gz file, we construct a FAISS index that we can query for the nearest neighbors of a given query embedding. We perform index construction in a multi-threaded manner to speed up the process. Upon constructing a FAISS index for all the json.gz files, we iterate over each query paper, search for the top 100 relevant papers using the SPECTER embeddings in *each* FAISS index, and then finally merge the results to get the top 1000 papers for each query paper.

## D.5 Comparative analysis of the computational costs

We compare the costs of different LLMs for both stages in Table 11.

**Ranking:** We explore two types of LLM-based reranking mechanisms: permutation and debate ranking. For $n$ query papers (=500 for our RollingEval datasets) and top-$k$ candidates retrieved from S2 per query paper ($k$=100 in our experiments), permutation ranking would require $n$ API calls, whereas debate ranking would require $n * k$ API calls. Debate ranking needs more API calls as it involves one additional API call per candidate paper to generate the citation probability score and reasoning. Therefore, there are $k$ additional API calls per query paper compared to permutation ranking, where we prompt the LLM to directly rank

---

[19]Code will be released at github.com

[20]https://huggingface.co/spaces/evaluate-metric/rouge Since it is a known issue in the NLG community of different implementations producing different results, we stick to evaluate==0.4.0 for reporting all the results, reproducing the ROUGE scores for baselines from Multi-XScience model outputs.

[21]https://spacy.io/usage/linguistic-features

[22]https://app.endpoints.anyscale.com/

[23]https://platform.openai.com/docs/guides/gpt

[24]https://api.semanticscholar.org/api-docs/graph

[25]https://api.semanticscholar.org/api-docs/recommendations

relevance for all the candidate papers. We refer the reader to Figure 15 for The exact prompt used for debate ranking.

**Generation:** There was only one request per query abstract in the RollingEval dataset, so 500 requests in total for each experiment (as $n = 500$ in RollingEval). The table below summarizes the API analysis for the two stages of the pipeline for the RollingEval experiments.

| Experiment | Method | Requests | Tokens | Cost |
|---|---|---|---|---|
| Ranking | GPT-4 Permutation Reranking | 500 | ~20M input + ~0.25M output tokens | $50 |
| | Llama-3.1 Debate Ranking (w/o attribution) | 500 x 100 | ~33M input + ~0.25M output tokens | $0 |
| | Llama-3.1 Debate Ranking (w/ attribution) | 500 x 100 | ~33M input + ~15M output tokens | $0 |
| Generation | Llama 2 70B (using Anyscale Endpoint) | 500 | ~0.75M input + ~0.15M output tokens | $3.84 |
| | GPT-3.5-turbo | 500 | ~0.75M input + ~0.15M output tokens | $4.2 |
| | GPT-4 | 500 | ~0.75M input + ~0.15M output tokens | $22 |
| | GPT-4 (Plan) | 500 | ~0.75M input + ~0.15M output tokens | $25 |

Table 11: Computational costs for different experiments on the RollingEval dataset. Costs for generation experiments on the Multi-XScience are approximately 10 times that of the RollingEval dataset.

---

**Abstract of Multi-XScience paper (Lu et al., 2020)**

**Reference @cite__1:** Multi-document summarization is a challenging task for which there exists little large-scale datasets. We propose Multi-XScience, a large-scale multi-document summarization dataset created from scientific articles. MultiXScience introduces a challenging multi-document summarization task: writing the related-work section of a paper based on its abstract and the articles it references. Our work is inspired by extreme summarization, a dataset construction protocol that favours abstractive modeling approaches. Descriptive statistics and empirical results—using several state-of-the-art models trained on the MultiXScience dataset—reveal that Multi-XScience is well suited for abstractive models.

**Abstract of Extractive and Abstractive Summarization paper (Pilault et al., 2020)**

**Reference @cite__2:** We present a method to produce abstractive summaries of long documents that exceed several thousand words via neural abstractive summarization. We perform a simple extractive step before generating a summary, which is then used to condition the transformer language model on relevant information before being tasked with generating a summary. We show that this extractive step significantly improves summarization results. We also show that this approach produces more abstractive summaries compared to prior work that employs a copy mechanism while still achieving higher rouge scores. Note: The abstract above was not written by the authors, it was generated by one of the models presented in this paper.

**Abstract of Galactica paper (Taylor et al., 2022)**

**Reference @cite__3:** Information overload is a major obstacle to scientific progress. The explosive growth in scientific literature and data has made it ever harder to discover useful insights in a large mass of information. Today scientific knowledge is accessed through search engines, but they are unable to organize scientific knowledge alone. In this paper we introduce Galactica: a large language model that can store, combine and reason about scientific knowledge. We train on a large scientific corpus of papers, reference material, knowledge bases and many other sources. We outperform existing models on a range of scientific tasks. On technical knowledge probes such as LaTeX equations, Galactica outperforms the latest GPT-3 by 68.2% versus 49.0%. Galactica also performs well on reasoning, outperforming Chinchilla on mathematical MMLU by 41.3% to 35.7%, and PaLM 540B on MATH with a score of 20.4% versus 8.8%. It also sets a new state-of-the-art on downstream tasks such as PubMedQA and MedMCQA dev of 77.6% and 52.9%. And despite not being trained on a general corpus, Galactica outperforms BLOOM and OPT-175B on BIG-bench. We believe these results demonstrate the potential for language models as a new interface for science. We open source the model for the benefit of the scientific community.

**Plan for Table 13**

Please generate 5 sentences in 120 words. Cite @cite__1 at line 1, 3 and 5.

**Plan for Table 13**

Please generate 5 sentences in 120 words. Cite @cite__1 at line 1 and 3. Cite @cite__2 at line 2 and 5. Cite @cite__3 at line 4 and 5.

Table 12: Abstracts of papers which are reference citations in Tables 13 and 14.

**Evaluate better Literature Review!**

Please provide ranks for model generated related work (1 best - you can also select both responses as 1s or 2s). Also mark hallucinations.

Instructions (click to expand) ◀

| Abstract | Model A | Rank |
|---|---|---|

Paper 3

Visual question answering (VQA) systems are emerging from a desire to empower users to ask any natural language question about visual content and receive a valid answer in response. However, close examination of the VQA problem reveals an unavoidable, entangled problem that multiple humans may or may not always agree on a single answer to a visual question. We train a model to automatically predict from a visual question whether a crowd would agree on a single answer. We then propose how to exploit this system in a novel application to efficiently allocate human effort to collect answers to visual questions. Specifically, we propose a crowdsourcing system that automatically solicits fewer human responses when answer agreement is expected and more human responses when answer disagreement is expected. Our system improves upon existing crowdsourcing systems, typically eliminating at least 20 of human effort with no loss to the information collected from the crowd.

Reference

Reference @cite_16: Foreground object segmentation is a critical step for many image analysis tasks. While automated methods can produce high-quality results, their failures disappoint users in need of practical solutions. We propose a resource allocation framework for predicting how best to allocate a fixed budget of human annotation effort in order to collect higher quality segmentations for a given batch of images and automated methods. The framework is based on a proposed prediction module that estimates the quality of given algorithm-drawn segmentations. We demonstrate the value of the framework for two novel tasks related to "pulling the plug" on computer and human annotators. Specifically, we implement two systems that automatically decide, for a batch of images, when to replace 1) humans with computers to create coarse segmentations required to initialize segmentation tools and 2) computers with humans to create final, fine-grained segmentations. Experiments demonstrate the advantage of relying on a mix of human and computer efforts over relying on either resource alone for segmenting objects in three diverse datasets representing visible, phase contrast microscopy, and fluorescence microscopy images.

Reference @cite_13: We consider a finite-pool data categorization scenario which requires exhaustively classifying a given set of examples with a limited budget. We adopt a hybrid human-machine approach which blends automatic machine learning with human labeling across a tiered workforce composed of domain experts and crowd workers. To effectively achieve high-accuracy labels over the instances in the pool at minimal cost, we develop a novel approach

**Model A**

The focus on efficient resource allocation in visual tasks is not new. @cite_12 explored a similar concept in their work on interactive segmentation algorithms. They considered the tradeoff between accuracy and effort and used a learning approach to predict if a graph cuts segmentation will succeed based on the image's visual separability and foreground uncertainty. This idea of a hybrid approach involving both human and machine input is also seen in the work by @cite_13, where they proposed a decision-theoretic active learning model for a finite-pool data categorization scenario. They found that blending machine learning with human labeling across a tiered workforce led to high-accuracy labels at minimal cost. The concept of 'pulling the plug' on human and computer annotators for image segmentation tasks was introduced by @cite_16. Their framework estimated the quality of algorithm-drawn segmentations and decided when to replace humans with computers or vice versa. While these works have significantly contributed to the field, our approach uniquely deals with the entangled problem in VQA systems and efficiently allocates human effort to collect answers to visual questions.

Rank: [ 1 ] [ 2 ]

Hallucination: ( ) True ( ) False

**Model B**

Previous work in the domain of resource allocation in image analysis has explored automated methods for efficiently integrating human and computer efforts (@cite_16). Similarly, our work on visual question answering (VQA) systems also emphasizes the efficient allocation of human effort. However, instead of improving segmentation quality, our focus is on predicting answer agreement within a crowd for a given visual question. Other research has adopted a hybrid human-machine approach for data categorization, leveraging decision-theoretic active learning to achieve high-accuracy labels at minimal cost (@cite_13). This shares similarities with our proposed crowdsourcing system, but it does not specifically address the problem of answer disagreement. Finally, research on the impact of manual annotation modes on segmentation algorithm accuracy and effort (@cite_12) parallels our work on VQA systems in terms of optimizing resource allocation, but differs significantly in the application domain.

Rank: [ 1 ] [ 2 ]

Hallucination: ( ) True ( ) False

[ Skip ] [ Submit ]

Use via API ⚡ · Built with Gradio 🟠

Figure 8: Interface of our human evaluation setup.

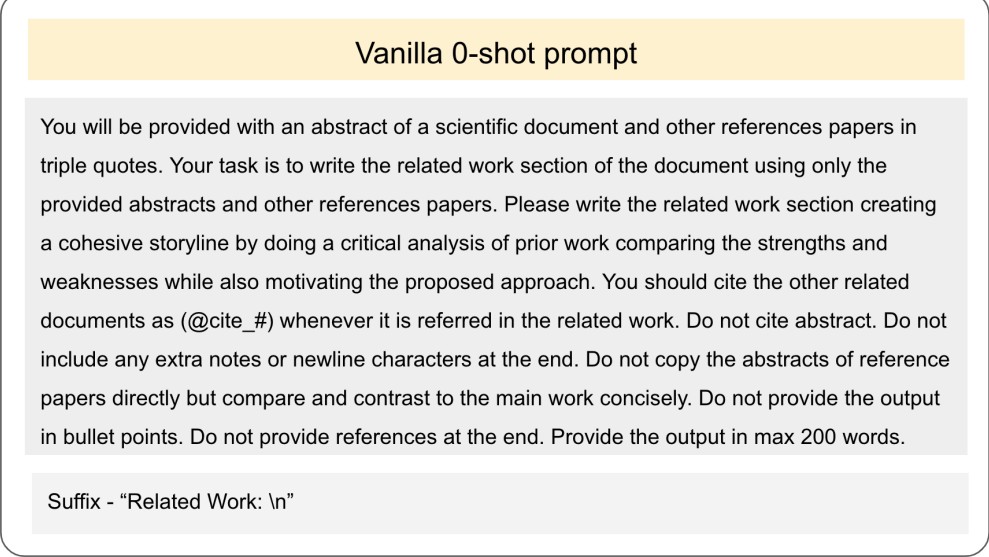

**Vanilla 0-shot prompt**

You will be provided with an abstract of a scientific document and other references papers in triple quotes. Your task is to write the related work section of the document using only the provided abstracts and other references papers. Please write the related work section creating a cohesive storyline by doing a critical analysis of prior work comparing the strengths and weaknesses while also motivating the proposed approach. You should cite the other related documents as (@cite_#) whenever it is referred in the related work. Do not cite abstract. Do not include any extra notes or newline characters at the end. Do not copy the abstracts of reference papers directly but compare and contrast to the main work concisely. Do not provide the output in bullet points. Do not provide references at the end. Provide the output in max 200 words.

Suffix - "Related Work: \n"

Figure 9: Prompt used for Vanilla 0-shot generation.

**GPT-4 0-shot**

The field of multi-document summarization has been a challenging one due to the lack of large-scale datasets (@cite_1). One significant attempt to address this gap is the Multi-XScience dataset, a large-scale collection of scientific articles designed to simulate the task of writing the related-work section of a research paper (@cite_1). This approach was inspired by extreme summarization, a methodology that promotes abstractive modeling approaches. However, the current study offers a novel strategy for literature review generation using large language models (LLMs), which goes beyond the scope of the Multi-XScience dataset. The proposed method involves the creation of an intermediate plan, which is then used to generate the actual text. Empirical evidence suggests that this approach improves the quality of the generated literature reviews compared to the conventional zero-shot generation. Additionally, this work expands the Multi-XScience corpus to include the full text of research papers, further enhancing the depth and breadth of information available for literature review generation. This study also ensures the validity of zero-shot experiments by creating a new test corpus from recent arXiv papers, thus avoiding test set contamination. This reflects a more comprehensive and rigorous approach to the generation of literature reviews using LLMs.

**GPT-3.5 Plan**

The task of generating literature reviews using large language models (LLMs) has gained attention recently. One approach is to condition the LLM on the abstracts of the papers to be cited and generate the literature review directly. However, the authors propose a novel strategy where they first generate a plan for the review and then use it to generate the actual text. This intermediate planning step is found to improve the quality of the generated literature reviews compared to vanilla zero-shot generation. The authors extend the Multi-XScience corpus, previously containing only abstracts, to include the full text of research papers. Additionally, they create a new test corpus consisting of recent arXiv papers, ensuring that their zero-shot experiments are not affected by contamination from the test set. The Multi-XScience dataset, used in their work, is particularly suitable for abstractive models due to its challenging multi-document summarization task (@cite_1). The empirical results from several state-of-the-art models trained on this dataset demonstrate its effectiveness for abstractive modeling approaches.

**GPT-3.5 0-shot**

Prior work in the field of generating literature reviews has primarily focused on the use of large language models (LLMs) and multi-document summarization. Multi-document summarization is a challenging task due to the lack of large-scale datasets (@cite_1). The Multi-XScience dataset was proposed as a solution to this problem, specifically designed for generating the related-work section of a paper based on its abstract and the articles it references. This dataset introduced a challenging multidocument summarization task that favors abstractive modeling approaches (@cite_1).

Our work builds upon these prior efforts by exploring the zero-shot abilities of recent LLMs for the task of generating literature reviews. We propose a novel strategy that involves generating an intermediate plan for the review, which is then used to generate the actual text. While modern LLMs can generate literature reviews without such intermediate plans, our empirical study shows that the use of these plans improves the quality of the generated reviews compared to vanilla zero-shot generation.

In addition to extending the Multi-XScience corpus to include the full text of research papers, we also create a new test corpus consisting of recent arXiv papers. This allows us to ensure that our zero-shot experiments are not affected by test set contamination.

The prior work on multi-document summarization and the Multi-XScience dataset provides a valuable foundation for our study. However, they do not specifically address the use of intermediate plans for literature review generation, which is a key contribution of our work. By incorporating these intermediate plans, we aim to enhance the quality and coherence of the generated literature reviews.

**Llama 2 Plan**

The task of generating literature reviews for scientific research papers has gained significant attention in recent years, with a focus on using large language models (LLMs) for zero-shot generation. Inspired by extreme summarization, Multi-XScience was introduced as a large-scale dataset for multi-document summarization, including the challenging task of writing related-work sections based on abstracts and referenced articles. Our work builds upon this foundation by exploring the effectiveness of intermediate plans for literature review generation. Our empirical study shows that these plans improve the quality of generated literature reviews, and we contribute to the extension of the Multi-XScience corpus to include full-text papers. Additionally, we create a new test corpus to ensure zero-shot experiments are free from test set contamination. (@cite_1)

**Llama 2 0-shot**

In recent years, there has been growing interest in using large language models (LLMs) for zero-shot literature review generation. However, most studies have relied on training or prompting LLMs to condition on all abstracts of papers to be cited, which can be limiting in terms of quality and efficiency. In contrast, our proposed approach involves generating a plan for the literature review and then using it to generate the actual text. We evaluate the effectiveness of this approach using a novel extended version of the Multi-XScience corpus, which now includes the full text of research papers. Our results show that intermediate plans improve the quality of generated literature reviews over vanilla zero-shot generation. One of the key challenges in multi-document summarization is the lack of large-scale datasets. To address this, we have created a new test corpus consisting of recent arXiv papers with full content, ensuring that our zero-shot experiments are not contaminated by the release of the LLMs used in our study. Our work builds on the concept of extreme summarization, which favors abstractive modeling approaches. We have employed several state-of-the-art models trained on the Multi-XScience dataset to evaluate the effectiveness of our approach. The results show that our dataset is well-suited for abstractive models, providing a valuable resource for researchers working in this area. In summary, our contribution includes a novel approach to literature review generation using LLMs, an extended version of the Multi-XScience corpus, and a new test corpus. Our results demonstrate the effectiveness of our approach and highlight the potential of using LLMs for zero-shot literature review generation. References: * @cite_1: Multi-XScience: A Large-Scale Multi-Document Summarization Dataset. * @cite_2: Extreme Summarization: A Dataset Construction Protocol for Abstractive Modeling.

Table 13: We show outputs of different models corresponding to the same plan and prompt as paragraph of our related work (by GPT-4 Plan). Llama 2 70B 0-shot has the tendency to hallucinate citation (in red) which is remedied in plan based generation. Abstract of reference paper shown in Table 12 while the main abstract is a slight modified version of our current abstract. We also find that the 0-shot results are more verbose, copying the exact text verbatim, while the generated literature review using plans are more representative of the way humans write literature review.

**GPT-4 Plan**

The concept of generating the related-work section from abstracts and referenced articles is not entirely new, as evidenced by the Multi-XScience dataset proposed by @cite_1. The approach of extending abstractive summarization to lengthy documents via an extractive step for conditioning, as described by @cite_2, aligns with the proposed methodology here. However, the strategy introduced in this study differentiates itself by creating an iterative plan prior to the generation of the review, a step not present in the work of @cite_1. The LLM used in this study also builds upon the capabilities of models like Galactica @cite_3, which have demonstrated exceptional performance on scientific tasks. The results of @cite_2 and @cite_3 support the potential of large language models in scientific literature review generation, while the proposed approach aims to further enhance the quality of these outputs through structured planning.

**GPT-4 0-shot**

The literature review generation task, a form of multi-document summarization, has been tackled through different strategies. @cite_1 proposed a dataset, MultiXScience, to foster abstractive approaches, while @cite_2 used an extractive step to enhance neural summarization. Our work complements these by introducing intermediate plans to improve generation quality. Meanwhile, @cite_3's Galactica demonstrated potential for language models in scientific tasks, a direction we explore with zero-shot experiments.

**GPT-3.5 Plan**

The task of generating a literature review using large language models (LLMs) has gained attention recently. One approach is to condition the LLM on the abstract and content of related papers to generate the review (@cite_1). However, it has been observed that generating an intermediate plan for the review improves its quality compared to vanilla zero-shot generation. This motivated the proposal and examination of a novel strategy where a plan is first generated and then used to generate the actual text (@cite_1). Another method to improve abstractive summarization is by performing an extractive step before generating the summary. This step conditions the transformer language model on relevant information, leading to better summarization results compared to prior work that employs a copy mechanism (@cite_2). Additionally, the use of large language models like Galactica has shown promising results in organizing and reasoning about scientific knowledge (@cite_3). Galactica outperforms existing models on technical knowledge probes and downstream tasks, demonstrating the potential for language models as a new interface for science (@cite_3).

**GPT-3.5 0-shot**

Prior work has explored the use of large language models (LLMs) for various natural language processing tasks. Multi-document summarization is one such task, and the Multi-XScience dataset (@cite_1) has been created specifically for this purpose. This dataset focuses on writing the related-work section of a scientific paper based on its abstract and the articles it references. The dataset construction protocol favors abstractive modeling approaches, making it suitable for LLMs. Another approach to abstractive summarization is presented in a paper (@cite_2), where the authors propose a method to generate summaries of long documents. They employ an extractive step to gather relevant information before conditioning the transformer language model for summary generation. This approach yields more abstractive summaries compared to prior work, while still achieving higher rouge scores. Furthermore, the concept of using LLMs for scientific knowledge organization and reasoning is introduced in the Galactica model (@cite_3). This model is trained on a large scientific corpus and outperforms existing models on various scientific tasks. It demonstrates superior performance on technical knowledge probes and reasoning tasks, surpassing GPT-3 and other models. Galactica also achieves state-of-the-art results on downstream tasks such as PubMedQA and MedMCQA, showcasing the potential of LLMs as a new interface for science. Based on these prior works, we propose a novel strategy for literature review generation using LLMs. We first generate a plan for the review and then use it to generate the actual text. While modern LLMs can generate literature reviews without intermediate plans, our empirical study indicates that these plans improve the quality of generated reviews. Additionally, we create a new test corpus consisting of recent arXiv papers, ensuring that our zero-shot experiments are not affected by test set contamination.

**Llama 2 Plan**

In recent years, there has been a growing interest in using large language models (LLMs) for scientific literature review generation (@cite_1). One approach is to use a novel strategy that involves generating a plan for the review and then using it to generate the actual text (@cite_1). This approach has been shown to improve the quality of generated literature reviews over vanilla zero-shot generation (@cite_1). Another approach is to use a simple extractive step before generating a summary, which is then used to condition the transformer language model on relevant information (@cite_2). This approach has been shown to produce more abstractive summaries compared to prior work that employs a copy mechanism while still achieving higher ROUGE scores (@cite_2). Additionally, there have been efforts to train LLMs on large scientific corpora to improve their ability to store, combine, and reason about scientific knowledge (@cite_3). For example, Galactica, a large language model, has been trained on a range of scientific tasks and has outperformed existing models on technical knowledge probes, reasoning, and downstream tasks (@cite_3). These advancements in LLMs have the potential to revolutionize the way we interact with scientific literature and accelerate scientific progress.

**Llama 2 0-shot**

In this section, we discuss the related work in the field of scientific literature review generation using large language models (LLMs). Recent works have explored the use of LLMs for multi-document summarization tasks, such as generating the related work section of a paper based on its abstract and the articles it references (@cite_1). These works have shown that LLMs can produce high-quality summaries, but may suffer from the lack of relevant information and context (@cite_2). To address this issue, we propose a novel strategy for literature review generation that involves generating a plan for the review before generating the actual text. This approach has been shown to improve the quality of generated literature reviews over vanilla zero-shot generation (@cite_3). Additionally, we create a new test corpus consisting of recent arXiv papers to ensure that our zero-shot experiments do not suffer from test set contamination. Our work builds upon the recent advances in LLMs and their ability to store, combine, and reason about scientific knowledge (@cite_3). We believe that our approach has the potential to revolutionize the way scientific literature is reviewed and organized, and we open-source our model for the benefit of the scientific community. In summary, our related work demonstrates the potential of LLMs for scientific literature review generation and highlights the importance of context and information in producing high-quality summaries. Our proposed approach aims to address these limitations and provide a more effective way of generating literature reviews using LLMs.

Table 14: We show outputs of different models corresponding to the reference cited abstracts and plan mentioned in Table 12 while the main abstract is a slightly modified version of our current abstract. In this example, though, we have all the citations covered by all the models, we can see GPT-4 (Plan) output to be concise and closely following the plan. Moreover, GPT-3.5 and Llama 0-shot outputs are excessively elaborate, making them unsuitable for inclusion in the literature review. Our findings indicate that while LLMs can help in certain aspects, the final output still heavily relies on inputs from a human researcher.

---

### Plan based prompt

You will be provided with an abstract of a scientific document and other references papers in triple quotes. Your task is to write the related work section of the document using only the provided abstracts and other references papers. Please write the related work section creating a cohesive storyline by doing a critical analysis of prior work comparing the strengths and weaknesses while also motivating the proposed approach. *You are also provided a plan mentioning the total number of lines and the citations to refer in different lines. You should cite the other related documents as (@cite_#) whenever it is referred in the related work.* Do not cite abstract. Do not include any extra notes or newline characters at the end. Do not copy the abstracts of reference papers directly but compare and contrast to the main work concisely. Do not provide the output in bullet points. Do not provide references at the end. Please follow the plan when generating sentences, especially the number of lines to generate. Provide the output in max 200 words.

Suffix - "Related Work: \n"

---

Figure 10: Prompt used for plan-based generation. Underlined text shows the variation compared to the vanilla 0-shot prompting, where the user provides a structure of the expected paragraph.

---

### Learned plan prompt

You will be provided with an abstract of a scientific document and other reference papers in triple quotes. Your task is to write the related work section of the document using only the provided abstracts and other reference papers. Please generate the related work creating a cohesive storyline by doing a critical analysis of prior work comparing the strengths and weaknesses while also motivating the new work. You should cite the other related documents as (@cite_#) whenever it is referred to in the related work. Do not cite abstract. Do not include any extra notes or newline characters at the end. Do not copy the abstracts of reference papers directly but compare and contrast to the main work concisely. Do not provide the output in bullet points. Do not provide references at the end. Provide the output in max 200 words. *You should first generate a plan, mentioning the total number of lines, words and the citations to refer to in different lines. You should follow this plan when generating sentences. \n Example: \n\n Plan: Generate the related work in [number] lines using max [number] words. Cite @cite_# on line [number]. Cite @cite_# on line [number].*\n

Suffix - "Related Work: \n"

---

Figure 11: Prompt used when the plan is learned during generation. The model first generates a plan of sentences and citations which it would then condition upon to generate the final related work text, which can be considered as an extension of CoT style thinking step by step.

---

### Sentence by sentence prompt

You are assisting a researcher to write a related work section of a paper sentence by sentence. *You will be provided with an abstract of the scientific document and raw draft of generated related work till now in triple quotes. Additionally, you will be provided with a reference paper if it has to be cited in the sentence. Your task is to write another 1 sentence for the related work section of the document or paraphrase the draft using only the abstract and other reference papers if provided. Initially, the raw draft would be empty.* Please complete the related work creating a cohesive storyline by doing a critical analysis of prior work comparing the strengths and weaknesses while also motivating the proposed approach. You should cite the other related documents as (@cite_#) only whenever it is referred to in the related work. Do not cite abstract. Do not include any extra notes or newline characters at the end. Do not copy the abstracts of reference papers directly but compare and contrast to the main work concisely. Do not provide the output in bullet points. Do not provide references at the end. Provide the output in max 200 words. *Provide the complete related work including the new sentence.*

Suffix - "Related Work: \n"

---

Figure 12: Prompt used for sentence-by-sentence generation. In this scenario, we prompt the model to generate one sentence for each citation individually.

---

### Per cite prompt

You will be provided with an abstract of a scientific document and other references paper in triple quotes. Your task is to write the related work section of the document using only the provided abstracts and other references papers. Please generate the related work creating a cohesive storyline by doing a critical analysis of prior work comparing the strengths and weaknesses while also motivating the new work. You should cite the other related documents as (@cite_#) whenever it is referred in the related work, comparing with the main paper. Do not cite abstract. Do not provide references at the end. Provide the output in 1-2 sentence.

Suffix - "Related Work: \n"

---

Figure 13: Prompt used for generating output per citation.

> ## Keyword summarization prompt
>
> You are a helpful research assistant who is helping with literature review of a research idea. You will be provided with an abstract of a scientific document. Your task is to summarize the abstract in max 5 keywords to search for related papers using API of academic search engine.
> ```Abstract: {abstract}```

Figure 14: Prompt used to summarize the research idea by LLM to search an academic engine

```
"""
You are a helpful research assistant who is helping with literature review of a research
    idea. Your task is to rank some papers based on their relevance to the query abstract.

## Instruction:
Given the query abstract:
<query_abstract>{query_abstract}</query_abstract>

Given the candidate reference paper abstract:
<candidate_paper_abstracts>{reference_papers}</candidate_paper_abstracts>

* Given the abstract of the candidate reference papers, provide me with a number between 0
    and 100 (upto two decimal places) that is proportional to the probability of a paper
    with the given query abstract including the candidate reference paper in its literature
    review.
* In addition to the probability, give me arguments for and against including this paper in
    the literature review.
* You must enclose your arguments for including the paper within <arguments_for> and </
    arguments_for> tags.
* You must enclose your arguments for including the paper within <arguments_against> and </
    arguments_against> tags.
* Extract relevant sentences from the candidate paper abstract to support your arguments.
* Put the extracted sentences in quotes.
* You can use the information in other candidate papers when generating the arguments for a
    candidate paper.
* You must enclose your score within <probability> and </probability> tags.
* Generate the arguments first then the probability score.
* Generate arguments and probabitlity for each paper separately.
* Do not generate anything else apart from the probability and the arguments.
* Follow this process even if a candidate paper happens to be identical or near-perfect
    match to the query abstract.

### Response Format for each paper:
<arguments_for>
[Paper ID]: [Reason for including the paper]
Extracted Sentences: "Sentence 1", "Sentence 2", ...
</arguments_for>
<arguments_against>
[Paper ID]: [Reason for not including the paper]
Extracted Sentences: "Sentence 1", "Sentence 2", ...
</arguments_against>
<probability>
[Paper ID]: [Final Probability Score Based on the Arguments]
</probability>

### Your Response:
"""
```

Figure 15: Prompt used for Debate Ranking.

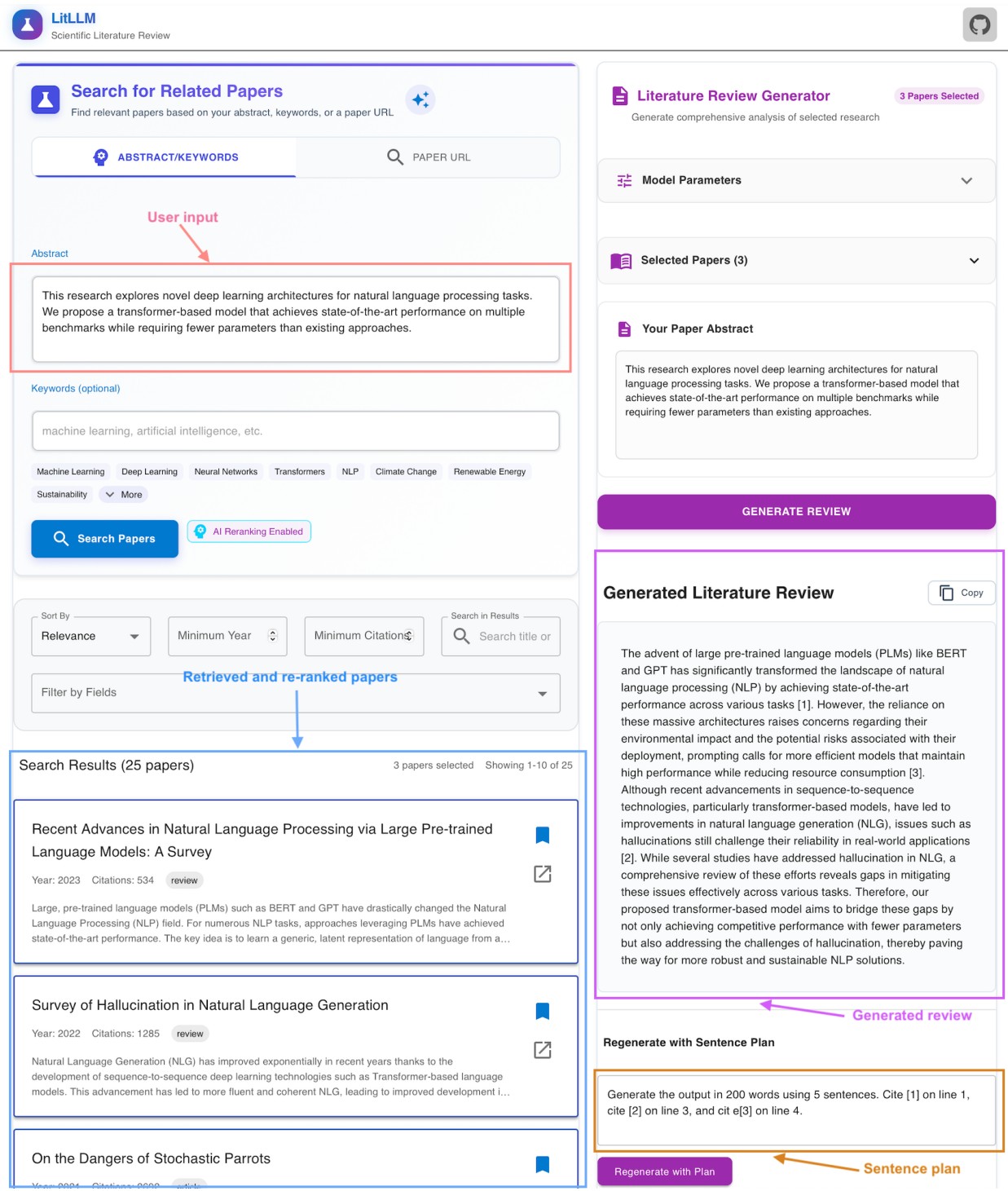

Figure 16: LitLLM interface Agarwal et al. (2024). Our system works on the Retrieval Augmented Generation (RAG) principle to generate the literature review grounded in retrieved relevant papers. The user needs to provide the abstract in the textbox (in purple) and press send to get the generated related work (in red). First, the abstract is summarized into keywords, which are used to query a search engine. Retrieved results are re-ranked (in blue) using an LLM, which is then used as context to generate the related work. Users could also provide a sentence plan (in green) according to their preference to generate a concise, readily usable literature review.

