# OpenReview forum: "LitLLMs, LLMs for Literature Review: Are we there yet?"
_TMLR — Accepted by TMLR_

### Review · Reviewer_LXGn · 2024-12-23

**Summary Of Contributions:**

This paper systematically studies the ability of Large Language Models (LLM) to write the related work section of an ML research paper given the abstract as a query. The authors carefully divide the complex and difficult problem into smaller subproblems, namely retrieving related papers, re-ranking the related papers into an order of relevance, generating a plan, and generating the related work section. The authors methodically show that, given specific instruction prompts, LLMs can solve these well-defined subproblems and help write the overall related work section. Moreover, the proposed pipeline supports human-in-the-loop at several stages, ensuring the writer can interact with the LLM and improve the generated response. The authors also collect two small-scale evaluation benchmarks from arXiv and promise the availability of both datasets and code in the future.

**Audience:**

Yes

**Claims And Evidence:**

Yes

**Requested Changes:**

Please refer to the weaknesses section for requested changes. Moreover, I also notice a few typos, such as 'keword' in section 7, please proofread the paper to address those.

**Strengths And Weaknesses:**

Overall, the paper does a fantastic job of addressing and dividing a complex problem into well-defined small-scale subproblems. The main strengths of this work are as follows:

(1) This is one of the few existing works to methodically evaluate the ability of LLMs to write a related work section for an ML paper. The division of this harder problem into easier subproblems enables us to analyze where the LLMs do well and where more manual intervention is required. For example, the authors clearly state that both the keyword-based and the ensembling-based paper retrieval techniques result in low coverage of retrieved papers. However, given the correct set of related papers and a clear, precise plan, the LLMs write the final section well.

(2) The pipeline is designed to enable human interaction in different stages. For example, a human can edit the list of retrieved papers or provide a manual plan to improve the quality of LLM’s response.

(3) The authors have experimented with various methods to solve each proposed subproblem. For example, (a) keyword and ensembling-based methods for paper retrieval, (b) Instructional permutation generation, SPECTER2 embeddings and Debate Ranking with Attribution for paper re-ranking, (c) Prompted plan, Per cite and sentence by sentence for plan generation, and (d) various extractive, abstractive, open- and closed-sourced LLMs for the final related work section writing. For each stage, the authors discuss the pros and cons of each technique.

(4) The authors propose a new evaluation benchmark for the task and promise it to be open-sourced.

(5) The main paper and supplementary provide multiple examples of the final outputs and the prompts used, which can be useful for future researchers deploying a similar system.

The primary weaknesses of the paper are as follows:

(1) One of the primary limitations of this work is the way to retrieve related papers. As the authors correctly say, the keywords do not represent the nuanced methodology presented by the paper. For example, the keywords from the popular CLIP paper can be “contrastive learning,” “vision-language pretraining,” “image classification,” and “dual encoders,” none of which can represent the significant contributions of the paper. Moreover, there are thousands of papers on each of these keywords, so finding the most relevant papers seems impossible. While the embedding-based strategy slightly improves the results, the best coverage achieved by the authors is only 9.8%. Hence, I request the authors to showcase if it is possible to get higher coverage when using the entire paper as the query or other potential methods to get more accurately retrieved papers.

(2) In section 3.4, the ground truth is all the cited references in the paper under query. However, many of the cited references are to add context from other work domains or to detail the implementation steps. Hence, all the cited references should not be treated as ground truth for related paper retrieval. Instead, only the papers cited in the related work section of the current paper may be a better choice for ground truth. If I understand it wrong, please correct me and clarify it in section 3.4.

(3) While the paper is generally well-written, the three different plan generation techniques in section 4.1 are confusing. The authors should make it more straightforward to perceive with smaller examples. Please also highlight the portion of the instructions (Figures 10, 11, 12, 13) that differs from the proposed approaches.

(4) For the example responses (Tables 11, 12, 13), please highlight where the model does a good job and where it hallucinates. Since the examples are very long, it is difficult for the reader to consume the takeaways. Moreover, please provide a few explicit examples where the model fails and analyze the possible reasons. Such an error analysis would be helpful for future work.

(5) Lastly, the fonts in Figures 2 and 3 are extremely small and hard to read. Please improve these two figures by utilizing the white space.

(6) The authors primarily use the ROUGE score to evaluate the generated related work section. However, since ROUGE analyzes token-level similarity, this is a terrible metric for such a long-form text comparison. On the other hand, human evaluation is expensive and subjective, depending on the expertise of the evaluator. I would encourage the authors to explore the possibility of using other semantic meaning-preserving metrics that have been used for long-form text comparison.

---

> ### Author Response · Authors · 2025-01-16
> **Response to Reviewer LXGn's comments**
>
> We sincerely thank the reviewer for their thoughtful comments and constructive feedback.
>
> **W1:**
> We had explored using full-length candidate papers in the ranking stage. However, our preliminary experiments on a smaller set of 100 query papers with 40 candidate papers each incurred high latency and inference costs without any gains in performance. The table below compares the two methods (on a subset of RollingEval-Dec dataset).
>
> | Method | Normalized Recall@20 | Precision@20 |
> | --- | --- | --- |
> | Debate Ranking (Full PDF) |  30.1 | 3.2 |
> | Debate Ranking (abstract only) | 48.9 | 4.6 |
>
> We hypothesize that this might be due to a limited long-context understanding of the current SOTA LLMs, as we noticed the LLM was picking up on irrelevant information from the first one or two pages, and did not retrieve any meaningful details from furthur down the PDF. We believe LLMs will improve in the future on this aspect. As you also suggested in the W2, we expect that the coverage will increase if we consider the citations only in the related work section.
>
> **W2:**
> This is an excellent observation! Indeed this effect explains why our coverage statistics are relatively low, i.e. some of the papers in the reference list are not actually found in the literature review. As future work we will suggest that follow-up work segregate the ground truth citations according to where they are found in the query paper. We had run initial trials to perform this kind of filtering; however, we found that the reliability of the obvious keyword based section identification approach would require significant additional human labeling to reliably construct this variation of the dataset in which one has isolated the citations only in the related work section.
>
> **W3:** Thank you for the suggestion. We updated the relevant figures 10,11 and 12  to highlight the differences in instructions (as underlined text) for different prompt generation techniques and also updated the corresponding text accordingly.
>
> **W4:** As suggested, we have updated the response tables to be more informative by highlighting the hallucinations and also discussed the shortcoming and advantages in Table 13 and 14 .
>
> **W5:** We have now updated the alignment of Figures 2 and 3 to make the font more legible. You can find them on page 7 and page 8 of the updated PDF, respectively.
>
> **W6:**
> We agree that ROUGE scores fail to capture the semantic correctness of texts. We report them to be consistent with previous work.
> Additionally, we have now added BERTScore and LLaMA-3-Eval results in Table 4 that have been proven to show a higher degree of alignment with human preferences than ROUGE scores.We also provide human evaluation results in Figure 5, where our instructions to the users in the human study regarding hallucinations included an emphasis on factual correctness. Human evaluators were also asked to consider fluency while grading the LLM responses.
> We report coverage in Table 6, which measures the percentage of model outputs with the same number of citations as ground truth. We also evaluate the plan-following capabilities in Table 5.

---

> > ### Comment · Reviewer_LXGn · 2025-02-03
> > **Thanks Authors for their Response**
> >
> > Thanks to the authors for addressing the weaknesses.
> >
> > > Low coverage of related papers.
> >
> > I still think increasing the coverage for finding related works is very important; the authors discussed the existing issues in the first two points, and I also requested that they incorporate this discussion along with the table (full PDF vs. abstract) in the final manuscript.
> >
> > > Improving figures and writing.
> >
> > Thanks for addressing my comments.
> >
> > >  BERTScore and LLaMA-3-Eval results in Table 4.
> >
> > Thanks for including these metrics. One additional suggestion from my end: since Tables 3 and 4 are large, please highlight / bold the best-performing methods and explain the main takeaway / summary of the table in the caption. This will help readers quickly consume the tables.
> >
> > Overall, I find the work a good contribution and suggest a 'weak accept.' I also request the authors release the evaluation benchmark and the RollingEval framework.

---

### Review · Reviewer_V4SW · 2025-01-07

**Summary Of Contributions:**

- The paper studies the zero-shot abilities of recent LLMs to support literature reviews by splitting the task into retrieval and review generation.
- Authors propose a multi-step pipeline: (1) prompt-based keyword extraction, (2) multi-engine retrieval, (3) LLM-driven re-ranking, and (4) plan-based text generation.
- Authors introduce new rolling evaluation datasets from arXiv to reduce training/test overlap with current models and measure retrieval coverage and generation quality.

**Audience:**

Yes

**Claims And Evidence:**

Yes

**Requested Changes:**

- Clarify how you will address ROUGE’s limitations in capturing factual correctness and fluency.
- Provide a detailed rationale for restricting citations to specific lines.
- Test newer LLMs (e.g., GPT-4 o1/o1-mini and Llama-3.1-8B) to show whether the approach generalizes well to more advanced models.

**Strengths And Weaknesses:**

## Strengths

- This work provides insight on tackling the time-intensive process of literature review generation with a clear task decomposition.
- Experiments across multiple LLMs (GPT-3.5, GPT-4, Llama 2, Code Llama) demonstrate gains from plan-based generation and re-ranking.
- Introduction of a rolling evaluation protocol to avoid test set contamination and a publicly available demo/interface.

## Weaknesses

- The LLMs tested (GPT-4, Llama 2, etc.) may already be behind the latest GPT variants (o1, o1-mini) and Llama-3.1-8B in capability. This raises the question of whether performance comparisons still hold for emerging SOTA models.

## Questions

- It seems that the paper evaluates its generation experiments by comparing the generated literature reviews to ground-truth text from the paper using the ROUGE metric. However, such n-gram-based scores (e.g., ROUGE) may not capture critical aspects like factual correctness or fluency. How do the authors address these limitations, and do they plan to supplement these rough metrics with other metrics?
- I am confused about the design of restricting models to mention citations on specific lines. Could the authors explain the motivation behind this design?

---

> ### Author Response · Authors · 2025-01-16
> **Response to Reviewer V4SW's comments**
>
> We appreciate the thoughtful review and the recognition of our contributions.
>
> **Regarding using emerging SOTA models:**
>
> We now have results for LLaMA-3.1-70b (Plan) and LLaMA-3.1-70b (No plan) in Table 3. We observe that it achieves comparable performance to GPT-4; however, newer LLMs like o1 and LLaMA-3.1 might also have data contamination issues for plan-based generation of related work on the RollingEval datasets and the current response period did not provide enough time to obtain a newer dataset and re-run all experiments. As a side note, closed-source models like o1 are quite expensive to run.
>
> **Regarding using additional metrics for evaluation:**
>
> We agree that ROUGE scores fail to capture the semantic correctness of texts. We report them to be consistent with previous work.
>
> Additionally, we have now added BERTScore and LLaMA-3-Eval results in Table 4 that have been proven to show a higher degree of alignment with human preferences than ROUGE scores.
>
> We also provide human evaluation results in Figure 5, where our instructions to the users in the human study regarding hallucinations included an emphasis on factual correctness. Human evaluators were also asked to consider fluency while grading the LLM responses.
> We also report coverage in Table 6, which measures the percentage of model outputs with the same number of citations as ground truth. We also evaluate the plan-following capabilities in Table 5.
>
> **Re rationale for restricting citations to specific lines:**
>
> As outlined in our paper, we intentionally adopted this approach to accommodate author preferences and potential publication constraints within the human-in-the-loop framework. These plans are designed and generated to allow user interaction and edits as needed. We view this as a form of scaffolding that helps mitigate hallucinations and enhance factual accuracy by providing a structured framework for related work and a general guideline for the section's length—essentially functioning as an optional feature.

---

> > ### Comment · Reviewer_V4SW · 2025-02-02
> >
> > > Regarding using emerging SOTA models:
> >
> > Thank you for including the LLaMA-3.1-70b experiment. It’s interesting to see that its performance is similar to that of Llama-2-Chat 70B.
> >
> > > Regarding using additional metrics for evaluation:
> >
> > Thank you for adding BERTScore, Llama-3-Eval and the human evaluation experiment. While I believe there is still room to develop better metrics and evaluation frameworks for benchmarking literature reviews generated by LLMs—given the task’s complexity in selecting, grouping, and comparing relevant literature—I am satisfied with the current evaluation presented in the paper.
> >
> > > Re rationale for restricting citations to specific lines:
> >
> > Thanks for the explanation.
> >
> > Additional Note: With new LLMs constantly emerging, creating a leakage-free benchmark for literature review generation remains a significant challenge. I believe the RollingEval framework should continue to be updated, as it can be a valuable resource for the community. It will be great if authors can release the dataset and the RollingEval framework.

---

### Review · Reviewer_Wncb · 2025-01-07

**Summary Of Contributions:**

This paper evaluates and improves the LLM capability in generating literature review. Authors decompose the literature review generation task into two key components: related work retrieval and review generation. The main contributions lies in:
1. A new dataset and rolling evaluation protocol using recent arXiv papers to assess zero-shot LLM performance while mitigating test contamination.
2. A hybrid search mechanism combining keyword-based and embedding-based retrieval, resulting in enhanced recall performance.
3. The proposed plan-based generation techniques for review generation reducing hallucinated references and improving coherence and citation coverage.

**Audience:**

Yes

**Broader Impact Concerns:**

As authors mentioned in Appendix A, there are legitimate concerns regarding the use of LLMs for generating literature reviews. These concerns include the potential for misuse, such as researchers relying excessively on LLMs without conducting thorough investigations into related work, which could lead to significant gaps in the literature coverage. Additionally, LLM-generated reviews may suffer from issues like hallucinations or the inclusion of non-existent references. Nevertheless, the exploration undertaken in this paper to be highly meaningful and a valuable contribution to the field.

**Claims And Evidence:**

Yes

**Requested Changes:**

I would recommend that the authors provide a comparative analysis of the computational costs (inference cost or API calls) associated with different methods, as well as across various stages of the proposed framework. This would offer valuable insights into the efficiency and scalability of the approach.

**Strengths And Weaknesses:**

### Strengths
1. Good performance. The proposed hybrid retrieval approach yields up to 30% improvement in recall compared to baselines, and the plan-based generation techniques reduces hallucinated references and improvs coherence and citation coverage.
2. Through rigorous analysis and well-designed human evaluation.
3. The use of rolling evaluations with recent datasets avoids data leakage, a critical aspect in LLM assessment.
4. Each pipeline components is well-evaluated separately.
### Weakness
1. I appreciate the novelty of the authors' approach in the Related Work section, where they employ LLM-generated literature reviews, aligning with the paper's theme. However, while reading this section, I found the logical flow to be somewhat disorganized. I would recommend restructuring it to enhance clarity, coherence, and conciseness, ensuring that the presentation of ideas is more streamlined and easier to follow.
2. Need a comparative analysis of the computational costs (inference cost or API calls).

---

> ### Author Response · Authors · 2025-01-16
> **Response to Reviewer Wncb**
>
> We thank the reviewer for acknowledging the novelty of our work and thoughtful comments.
>
> **Lack of coherence/logical flow in the Related Work section:**
> We have updated the related work section in the paper to improve the flow. Specifically, we now have two subheadings for Retrieval and Generation and have also added a discussion on relevant attribution methods. Kindly refer to Section 2 of the updated PDF.
>
> **Comparative analysis of different methods:**
> We updated the manuscript with comparative cost analysis in Appendix D.4. Please find the table below comparing costs of different LLMs for different stages:
>
> Ranking:
> We explore two types of LLM-based reranking mechanisms: permutation and debate ranking.
> If we have $n$ query papers (=500 for our RollingEval datasets) and choose top-$k$ candidates from S2 (=100 in our experiments), we need to make $n$ API calls for permutation ranking and $n*k$ requests for debate ranking.
>
> Generation:
> There was only one request per query abstract in the RollingEval dataset, so 500 requests in total for each experiment (as $n=500$ in RollingEval). The table below summarizes the API analysis for the two stages of the pipeline for the RollingEval experiments.
>
> | **Category**        | **Method**                              | **#Requests**  | **Tokens Consumed**                 | **Cost**  |
> |----------------------|-----------------------------------------|----------------|--------------------------------------|-----------|
> | **Ranking**          | GPT-4 Permutation Reranking            | 500            | ~20M input + ~0.25M output tokens   | $50       |
> |                      | LLaMA-3.1 Debate Ranking (w/o attribution) | 500 × 100      | ~33M input + ~0.25M output tokens   | $0        |
> |                      | LLaMA-3.1 Debate Ranking (w/ attribution) | 500 × 100      | ~33M input + ~15M output tokens     | $0        |
> | **Generation**       | LLaMA 2 70B (using Anyscale Endpoint)  | 500            | ~0.75M input + ~0.15M output tokens | $3.84     |
> |                      | GPT-3.5-turbo                          | 500            | ~0.75M input + ~0.15M output tokens | $4.2      |
> |                      | GPT-4                                  | 500            | ~0.75M input + ~0.15M output tokens | $22       |
> |                      | GPT-4 (Plan)                           | 500            | ~0.75M input + ~0.15M output tokens | $25       |
>
> **Regarding the broader impact concern:**
>
> We agree with the concerns of the reviewer, and we want to clarify again that our primary focus in this work was to gauge the ability of LLMs for writing literature reviews. Our findings indicate that while LLMs can help in certain aspects of the process, the final output still heavily relies on inputs from a human researcher. The issue of hallucinated nonexistent references strongly motivated our proposed two phase process, and our experiments indeed showed that this approach indeed significantly mitigates hallucinated references. It is our hope that this tool will expose researchers to a wider variety of related prior work than they might normally be able to identify using traditional research tools and if implemented in a working system we think that researchers should be strongly encouraged to actually read any papers that this kind of system has identified as being suitable for including in their literature review.

---

### Comment · Action_Editor_kiUU · 2025-01-08
**Author/Reviewer discussion**

Dear Authors,

Please take a moment to carefully review the feedback provided by the three reviewers. You have two weeks to engage in discussions with them before moving with the decision-making process.

Best regards,
AE

---

### Decision · Action_Editor_kiUU · 2025-02-18

**Recommendation:** Accept with minor revision

**Comment:**

(1) I currently cannot access the code. If you intend to release it, please replace the placeholder with the actual repository link.

(2) Reviewer Wncb pointed that "the inference incurs a relatively high cost". A bit more clarity on computational overhead and cost (e.g., how many prompts, tokens, approximate cost ranges, which is complementary to Section D.4) could help readers plan practical usage.

**Audience:**

The paper tackles practical aspects of using LLMs for literature review. This application is relevant to researchers who are pressed for time and looking for automated ways to discover and summarize relevant work. Readers working on research tools, agent, and information retrieval are especially likely to find value in the findings.

**Claims And Evidence:**

This submission, “LLMs for Literature Review: Are we there yet?”, investigates how LLMs can be used to assist authors in writing literature reviews. The paper decomposes the problem into two major sub-tasks: (1) retrieval of relevant papers and (2) generation of the related work text. The authors propose a pipeline that includes LLM-driven keyword extraction, multi-engine retrieval, LLM-based re-ranking (with an optional attribution step), and plan-based text generation. Reviewers are satisfied with the revision.

---

> ### Comment · Action_Editor_kiUU · 2025-03-25
> **Empty code repo**
>
> The code repo of https://github.com/LitLLM/litllms-for-literature-review-tmlr is empty.
>
> Please ensure it is updated.

---

> > ### Author Response · Authors · 2025-03-26
> > **Updated GitHub**
> >
> > Dear Action Editor,
> >
> > We have now populated the GitHub repo with the code along with usage instructions. We extend our sincere thanks to all the reviewers for their meaningful feedback on our work.